# Experimental validation of a modeling framework for upconversion enhancement in 1D-photonic crystals

Clarissa L. M. Hofmann [1,2✉], Stefan Fischer[3], Emil H. Eriksen[4], Benedikt Bläsi [1], Christian Reitz[5], Deniz Yazicioglu [1,6], Ian A. Howard [2,7], Bryce S. Richards [2,7] & Jan Christoph Goldschmidt [1]

Photonic structures can be designed to tailor luminescence properties of materials, which becomes particularly interesting for non-linear phenomena, such as photon upconversion. However, there is no adequate theoretical framework to optimize photonic structure designs for upconversion enhancement. Here, we present a comprehensive theoretical model describing photonic effects on upconversion and confirm the model's predictions by experimental realization of 1D-photonic upconverter devices with large statistics and parameter scans. The measured upconversion photoluminescence enhancement reaches 82 ± 24% of the simulated enhancement, in the mean of 2480 separate measurements, scanning the irradiance and the excitation wavelength on 40 different sample designs. Additionally, the trends expected from the modeled interaction of photonic energy density enhancement, local density of optical states and internal upconversion dynamics, are clearly validated in all experimentally performed parameter scans. Our simulation tool now opens the possibility of precisely designing photonic structure designs for various upconverting materials and applications.

[1] Fraunhofer Institute for Solar Energy Systems, Heidenhofstraße 2, 79110 Freiburg, Germany. [2] Institute of Microstructure Technology (IMT), Karlsruhe Institute of Technology, Hermann-von-Helmholtz-Platz 1, 76344 Eggenstein-Leopoldshafen, Germany. [3] Department of Materials Science and Engineering, Stanford University, 496 Lomita Mall, Stanford, CA 94305, USA. [4] Department of Physics and Astronomy, Aarhus University, Ny Munkegade 120, DK-8000 Aarhus, Denmark. [5] Institute of Nanotechnology (INT), Karlsruhe Nano Micro Facility, Karlsruhe Institute of Technology, Hermann-von-Helmholtz-Platz 1, 76344 Eggenstein-Leopoldshafen, Germany. [6] Laboratory for Nanotechnology, Institute of Micro Systems Technology – IMTEK, University of Freiburg, Georges-Köhler-Allee 103, 79110 Freiburg, Germany. [7] Light Technology Institute (LTI), Karlsruhe Institute of Technology, Engesserstrasse 13, 76131 Karlsruhe, Germany. ✉email: clarissa.hofmann@ise.fraunhofer.de

Photon upconversion (UC), the conversion of low-energy into higher-energy photons by use of lanthanide-doped materials, has been of rapidly growing interest in the fields of materials chemistry and physics within the past 50 years[1]. Extensive research has been done on understanding the theory of the UC process[1–4] and on material development, predominantly nanocrystals[5–15]. By now, UC is exploited in a broad range of applications ranging from bioimaging[13,16–20], theranostics[21–24], security[20,25,26], data storage[27], and data analysis[28] to photovoltaics[13,15,19,20,29–31]. The probability of an UC process increases non-linearly with increasing irradiance because two photons need to be absorbed in immediate vicinity in space and time[1]. For an application of UC in photovoltaics, the relatively low irradiance of the sun constitutes a challenge because it limits the UC efficiency. One approach to increase UC efficiency at low irradiances is to embed the upconverter into a photonic crystal. This photonic upconverter is then placed behind the solar cell (Fig. 1a).

Photonic structures that have been investigated for UC enhancement include regular[32–34] and inverse opal photonic crystals[35,36] also in combination with plasmonic effects[21,25,26,37–41], as well as 2D photonic crystals[42], waveguide structures[43], cavities[44,45], and multi-layer stacks[46]. The highest reported UC enhancement factors range from <30 for cavities[44] and opal structures[32–34], to three-to-four orders of magnitude for hybrid opal photonic structures combined with plasmonic resonances[25,37] and waveguide structures[43]. These results demonstrate the high

potential of photonic structure enhanced UC. A detailed overview can be found in the Supplementary Tables 1–4. Also detailed understanding of plasmonic enhancement effects are of major interest in various areas of application[47–49]. However, the theoretical understanding of how photonic effects influence UC and its implementation in simulation models is mostly lacking. Without this understanding, a proper photonic structure design optimization is not possible and the actual potential of the photonic structure cannot be fully exploited. Additionally, more complex structures are more sensitive to structural imperfections. Considering a given production accuracy, the same UC enhancement could be reached with a less complex structure[50], which is particularly relevant for industrial applications. Furthermore, the maximum enhancement factor that is reported in publications is mostly measured at one very distinct set of parameters (i.e. excitation wavelength and irradiance, incidence or detection angle etc.), thus not including statistics or the spectral width of the UC enhancement, which are very decisive parameters for some target applications, including photovoltaics. In addition, it is unclear if the reported UC enhancement predominantly stems from a photonic enhancement or simply an enhanced fraction of absorbed excitation light due to scattering, for example.

To fill these gaps, we have developed a comprehensive theoretical model, describing the influence of both photonic effects, the local energy density and modified local density of optical states (LDOS), on the internal UC dynamics of $Er^{3+}$ in the host crystal hexagonal $NaYF_4$ (refs. [50–52]). Additionally, the model

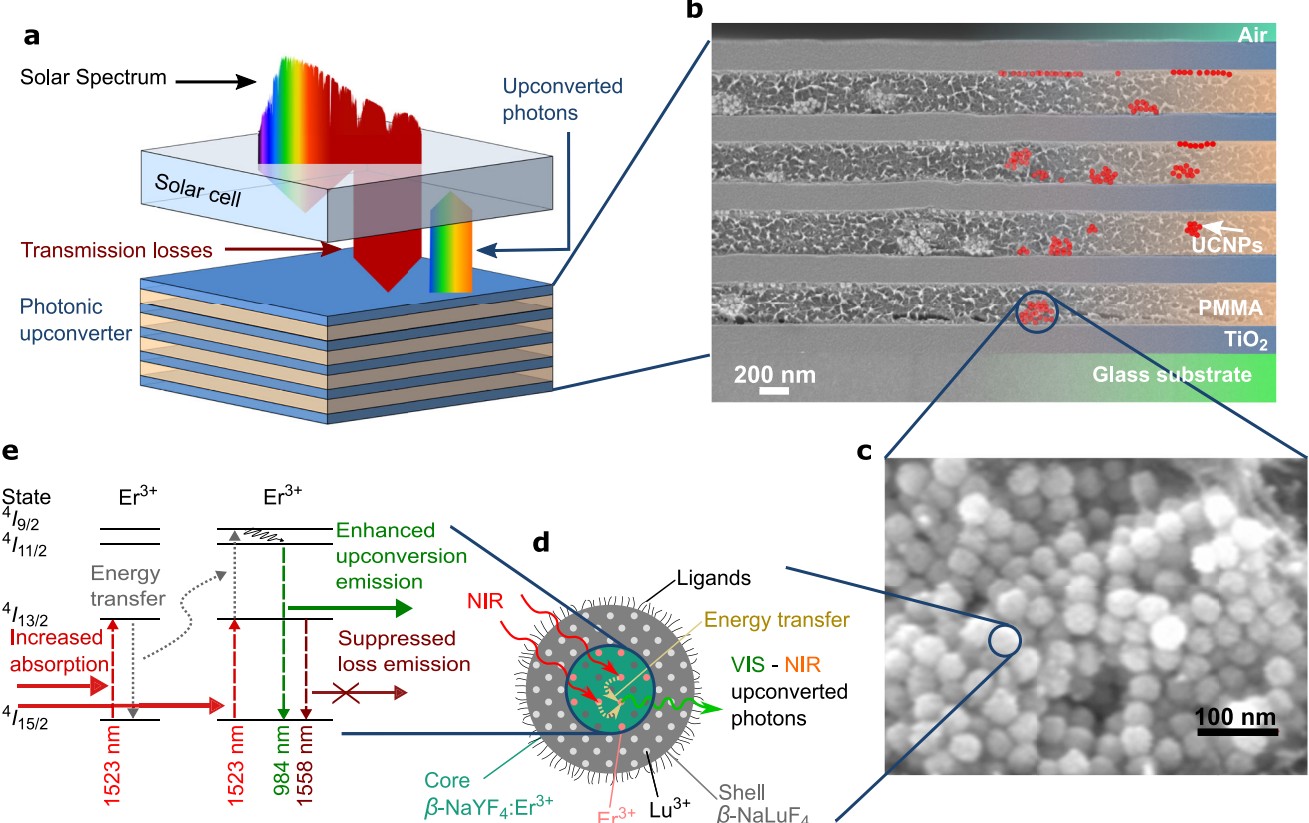

**Fig. 1 Motivation of the investigated photonic upconverter device. a** Approach of utilizing sub-bandgap photons for charge generation in a solar cell by a photonic upconverter on the rear side. **b** Scanning electron microscope (SEM) image of the realized 1D-photonic structure made of $TiO_2$ and PMMA with embedded upconverter nanoparticles (UCNPs). **c** SEM image of upconverter nanoparticles. **d** Schematics of core-shell upconverter nanoparticles of $NaYF_4$: $Er^{3+}$, converting near infrared (NIR) to NIR and up to visible (VIS) photons in the active core. The inert shell prevents losses due to surface quenching. **e** Energy levels in the upconverter $Er^{3+}$ and the upconversion (UC) process influenced by photonic effects of the surrounding structure: increased absorption due to a locally enhanced energy density, non-linearly increasing the probability of an energy transfer UC process, followed by UC emission from a higher level that can be enhanced due to a modified local density of optical states.

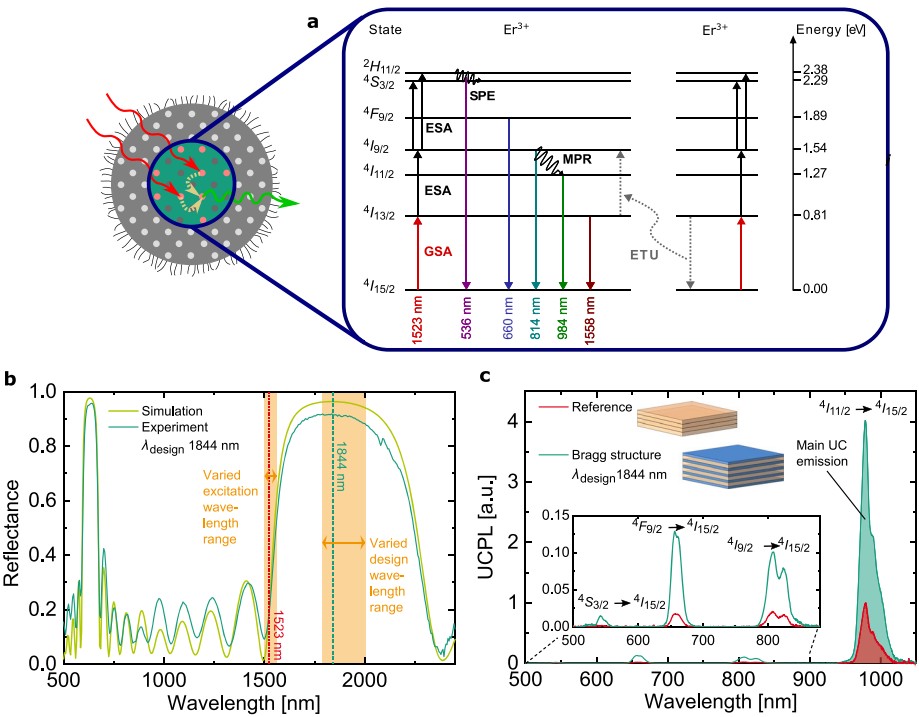

**Fig. 2 Design and upconversion photoluminescence (UCPL) of a Bragg structure. a** Energy level diagram of the first seven energy levels of $\beta$-NaYF$_4$:25% Er$^{3+}$, including the processes: ground- and excited-state absorption (GSA, ESA), multi-phonon relaxation (MPR), energy transfer upconversion (ETU) (one exemplary ETU process shown), and spontaneous emission (SPE). **b** Reflectance of a fabricated Bragg structure with the matched simulated reflectance at a design wavelength $\lambda_{design} = 1844$ nm. The 40 investigated sample designs range from $\lambda_{design}$ of 1784 nm to 2005 nm. For UCPL measurements, the excitation wavelength is varied from 1500 nm to 1560 nm. **c** Measured UCPL under 1523 nm excitation at 1.48 W cm$^{-2}$ irradiance using an integrating sphere to collect the integrated light from all angles. Due to the photonic effects on UC, in the Bragg structure, all UC emission is significantly enhanced. The relative enhancement of the main UCPL at 984 nm (UCPL$_{rel}$) in the Bragg structure compared to the reference is 4.1.

considers the effect of experimental production inaccuracies[50]. We choose to investigate a simple Bragg structure design that reveals the essential aspects and is also application relevant (it could be fabricated on an industrial scale). Another key advantage of a layer stack system is that it is possible to add many layers, so the overall volume of upconverter material on which the photonic structure acts can be large. The overall absorption can therefore be high, unlike in other systems, where high enhancements are confined to very small volumes. The structure we investigate consists of alternating quarter wave layers of TiO$_2$ and poly (methyl methacrylate) (PMMA) (Fig. 1b), containing custom-made hexagonal NaYF$_4$:25%Er$^{3+}$ ($\beta$-NaYF$_4$:25%Er$^{3+}$) core-shell upconverter nanoparticles (Fig. 1c–e). We validate the predictions of a comprehensive theoretical model by experimentally realizing these 1D-photonic upconverter devices in 40 different sample designs and by performing a large-scale parameter scan, investigating irradiance and excitation wavelength, compiling statistics of 2480 measurements.

## Results

**Optimization of photonic upconverter devices**. We fabricated optimized Bragg structures comprising of five TiO$_2$ layers and four intermediate layers of PMMA with embedded core-shell upconverter nanoparticles of $\beta$-NaYF$_4$:25%Er$^{3+}$, in the following referred to as active layers ("Methods"). The scanning electron microscope (SEM) cross-sectional image of a fabricated Bragg structure demonstrates the high layer quality and uniformity (Fig. 1b, see also Supplementary Fig. 8a). The upconverter nanoparticles mostly form small clusters within the PMMA layer or agglomerate at the layer surface. Nevertheless, the surface

roughness of the topmost layer of the displayed Bragg structure is only in the order of 10 nm (Supplementary Fig. 8b).

Figure 2a shows the first seven energy levels of Er$^{3+}$ in the host crystal $\beta$-NaYF$_4$. These processes are influenced by the surrounding of the Bragg structure, as motivated in Fig. 1e. For our study, the most important properties of the Bragg structure are the existence and position of the photonic bandgap, represented by the characteristic reflectance (Fig. 2b). The position of the reflectance peak, and therewith the first photonic bandgap, is determined by the design wavelength ($\lambda_{design}$) that defines the thickness $d_i = \lambda_{design}/4n_i$ of each layer $i$ with refractive index $n_i$. Fitting the measured to the simulated reflectance, we determined the exact design wavelength of each evaluated sample point ("Methods"). With the chosen sample designs, we can investigate the photonic effects, ranging from the expected maximum with an excitation at the photonic band edge, to an expected suppression.

To quantify the effect of a photonic structure on UC, we investigate the UC photoluminescence (UCPL) ("Methods") of a Bragg structure relative to its corresponding reference (Fig. 2c). As a reference, we choose one active layer on glass, featuring the same thickness as the sum of all active layers of the corresponding Bragg structure. The main UC emission around 984 nm contains 94% of the measured total UCPL, it stems from the electronic transition $^4I_{11/2}$ to $^4I_{15/2}$. The emission intensity, corresponding to this transition is enhanced in the Bragg structure by a factor of 4.1 due to the photonic effects. The $^4I_{9/2}$ to $^4I_{15/2}$ transition can be seen in the 814 nm UC emission, with an enhancement factor of 5.2. The 3-photon processes $^4F_{9/2}$ to $^4I_{15/2}$ at 660 nm and $^4S_{3/2}$ combined with $^4H_{11/2}$ to $^4I_{15/2}$ at 536 nm are enhanced by a factor of 7.3 and 8.9, respectively.

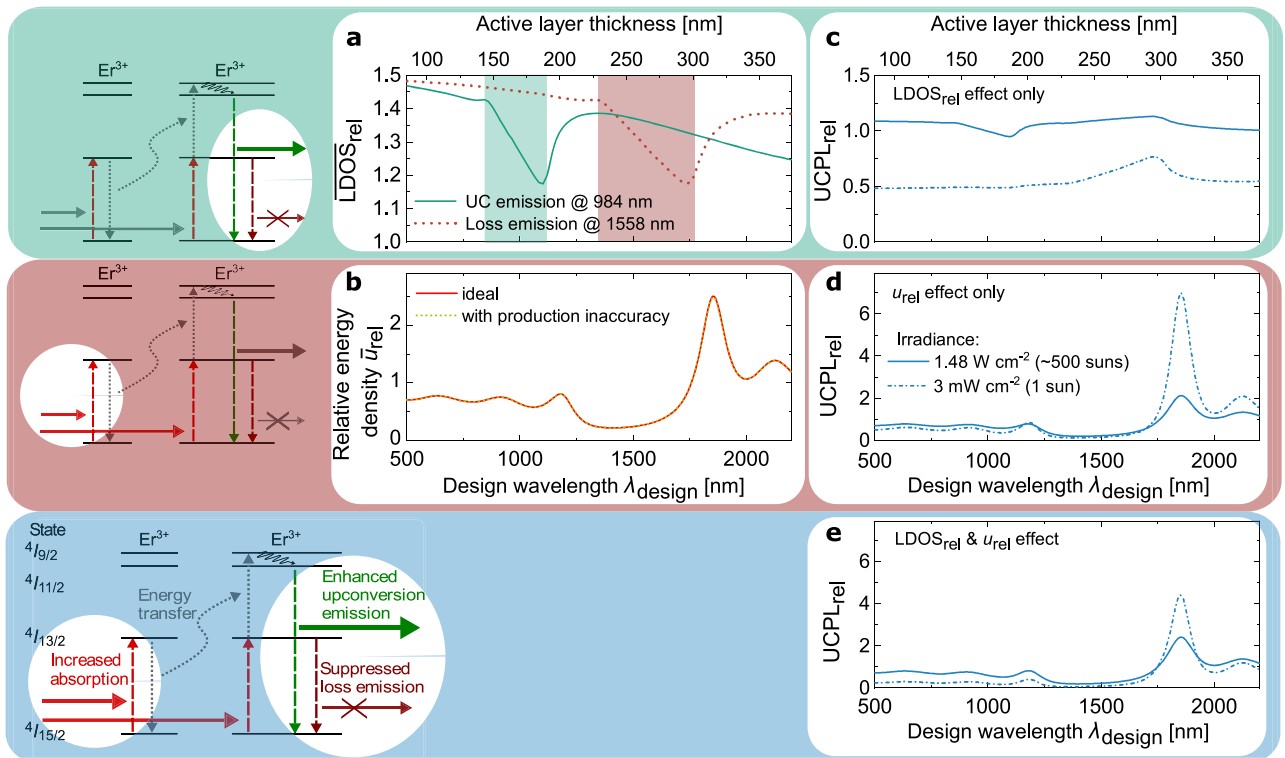

**Fig. 3 Photonic effects on upconversion (UC) as a function of the design wavelength $\lambda_{\text{design}}$. a** Average relative local density of optical states ($\overline{\text{LDOS}}_{\text{rel}}$) in the active layers of the Bragg structure for the main UC emission and main loss emission. **b** Average relative energy density ($\bar{u}_{\text{rel}}$) in the active layers of the Bragg structure for an excitation at 1523 nm for an ideal Bragg structure and the fabricated structure including measured production inaccuracies. **c–e** Relative UC photoluminescence (UCPL$_{\text{rel}}$) at 3 mW cm$^{-2}$ (1 sun), as well as at 1.48 W cm$^{-2}$ (~500 suns) irradiance as in experiment, only taking into account the LDOS effect (**c**), the effect of the relative energy density (**d**), and both effects (**e**). Under 1 sun illumination, the irradiance in the absorption range of the upconverter Er$^{3+}$ (1450 nm–1600 nm) is 3 mW cm$^{-2}$ (ref. [53]).

**Simulation of photonic effects on UC**. The effects of the Bragg structure, the change of the LDOS and the energy density, critically depend on the design wavelength $\lambda_{\text{design}}$. The induced changes can be expressed as relative values obtained by integrating over the active layers within the Bragg structure and dividing by the corresponding integral of the reference, yielding the average relative LDOS ($\overline{\text{LDOS}}_{\text{rel}}$) and the average relative energy density ($\bar{u}_{\text{rel}}$). From the locally resolved photonic effects, the change in the UC emission at 984 nm relative to the reference (UCPL$_{\text{rel}}$) is calculated ("Methods"). In Fig. 3, we investigate the change in UC emission due to photonic effects at two different irradiance levels that are relevant for the target application of photovoltaics: (i) at 1 sun illumination, where the irradiance in the absorption range of the upconverter Er$^{3+}$ between 1450 nm and 1600 nm is 3 mW cm$^{-2}$ (ref. [53]) and (ii) at 1.48 W cm$^{-2}$, corresponding to ~500 suns concentration, which is a typical regime for high-concentration photovoltaic systems[54].

Figure 3a shows $\overline{\text{LDOS}}_{\text{rel}}$ for the two most important spontaneous emissions, the main UC emission at 984 nm and the main loss emission at 1558 nm, the direct de-excitation of the first excited state (compare to Fig. 1e). In the region of $\lambda_{\text{design}}$, in which an emission lies inside the first photonic bandgap (highlighted regions in Fig. 3a), $\overline{\text{LDOS}}_{\text{rel}}$ is strongly reduced. However, as our reference consists only of the low refractive index material, there are more photonic states in the Bragg structure, and $\overline{\text{LDOS}}_{\text{rel}}$ is always above one. The net effect of the LDOS on UC efficiency is a complex, non-linear superposition of $\overline{\text{LDOS}}_{\text{rel}}$ of both emissions (Fig. 3c). An increase of $\overline{\text{LDOS}}_{\text{rel}}$ of the main UC emission at 984 nm linearly increases UC efficiency. However, an increase in $\overline{\text{LDOS}}_{\text{rel}}$ of the main loss emission at

1558 nm non-linearly decreases UC efficiency. At low irradiances, the increased probability of the 1558 nm loss emission is more relevant because the few available excited upconverter ions in the first excited state have a high probability to be de-excited again before an UC process can take place. For higher irradiances, this strong dependence on the probability of the 1558 nm loss emission loses its large impact as there are more excited upconverter ions available, which increases the probability that an UC process takes place before de-excitation.

The photonically modified energy density is very sensitive to structural imperfections[50]. Therefore, we include production inaccuracies in our simulation via a Monte-Carlo approach, using measured standard deviations of the layer thicknesses as input parameters ("Methods"). In Fig. 3b, $\bar{u}_{\text{rel}}$ is plotted for an excitation wavelength of 1523 nm, for an ideal Bragg structure, as well as the fabricated structure with layer production inaccuracies of 4.2 nm and 1.5 nm for the active and TiO$_2$ layers, respectively ("Methods"). The non-ideal $\bar{u}_{\text{rel}}$ is almost identical to the ideal $\bar{u}_{\text{rel}}$. This demonstrates that the production accuracy we reached in experiment is high enough that it does not diminish the photonic effects in the particular structure we are investigating. Because $\lambda_{\text{design}}$ determines the position of the reflectance peak for a broad region around $\lambda_{\text{design}}$ of 1500 nm, the excitation at 1523 nm falls into the photonic bandgap and is directly reflected. The peak enhancement is reached at $\lambda_{\text{design}} = 1855$ nm, when the excitation lies at the lower band edge.

Figure 3d shows UCPL$_{\text{rel}}$, only taking into account the effect of the relative energy density. The non-linear dependence of the UC process on the irradiance is well visible in this graph. At 3 mW cm$^{-2}$ irradiance, corresponding to 1 sun, the reference performs very

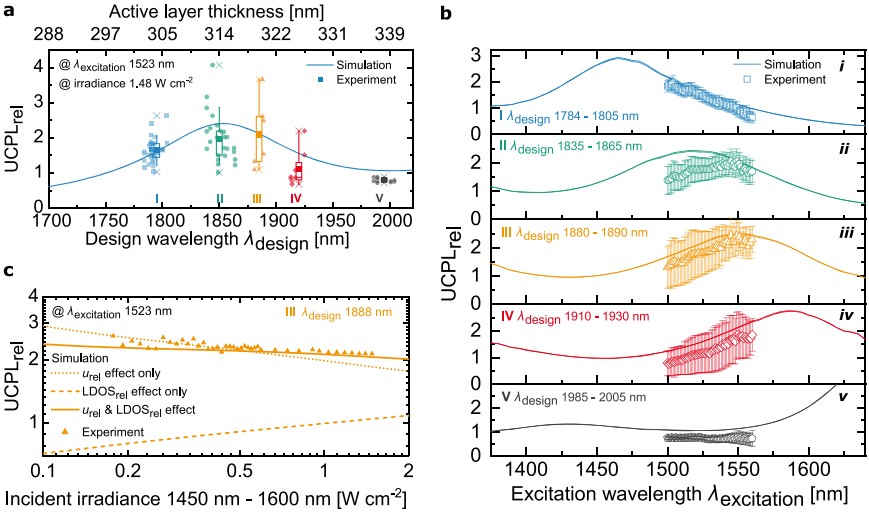

**Fig. 4 Effect of varied parameters on the relative upconversion photoluminescence (UCPL$_{rel}$)—comparison of simulation and experiment.** **a** We investigate the dependence of UCPL$_{rel}$ on the design wavelength $\lambda_{design}$ using 40 sample designs around the expected maximum UC enhancement, sorted into five groups (I–V) of similar $\lambda_{design}$. Two measurements of each investigated design are plotted, the boxes contain 50%, the whiskers 80% of the data points within each group. Point and horizontal line represent mean and median, respectively. **b** Scanning the excitation wavelength $\lambda_{excitation}$, the mean and standard deviation of UCPL$_{rel}$ within each group I–V is plotted. The applied irradiance in experiment lies between 1.57 W cm$^{-2}$ at $\lambda_{excitation} = 1500$ nm and 1.38 W cm$^{-2}$ at $\lambda_{excitation} = 1560$ nm. The simulation is plotted for the center $\lambda_{design}$ of each group at these two boundary irradiances. **c** Effect of varied irradiance for one sample design of group III, compared to simulation of UCPL$_{rel}$ including only one photonic effect, of the changed local energy density $u_{rel}$ or the modified local density of optical states LDOS$_{rel}$, or both effects. For all investigated parameter scans (**a**–**c**), the expected trends from simulation are clearly visible in experiment. In the mean of all 2480 measurements at separate parameter combinations, the experimentally measured UCPL$_{rel}$ divided by the simulated UCPL$_{rel}$ is 82 ± 24%, featuring a very good agreement. Source data for **a** and **b** (i–v) are provided as a Source Data file.

poorly because at this low irradiance very few ions are excited and the probability for an UC process to take place is very low. The energy density enhancement within the Bragg structure increases absorption and therefore strongly increases UC efficiency. At 1.48 W cm$^{-2}$ irradiance, corresponding to ~500 suns, the effect is less pronounced, because the additional energy density enhancement still enhances absorption, but also contributes to UC emissions from even higher excited states, thus reducing the benefit for an UC emission at 984 nm.

Finally, in Fig. 3e, both photonic effects are considered, revealing that the net effect on UC is a complex non-linear superposition of both. The shape of UCPL$_{rel}$ is very similar to Fig. 3d, showing that $\bar{u}_{rel}$ with its strongly pronounced maximum is decisive for optimizing $\lambda_{design}$. Nevertheless, the LDOS effect needs to be considered when regarding a particular irradiance. As demonstrated in Fig. 3c, the altered LDOS has a negative effect at low irradiances and a positive effect at high irradiances. In consequence, in Fig. 3e, compared to Fig. 3d, UCPL$_{rel}$ is reduced for the low irradiance and increased for the high irradiance by the LDOS effect. Thus, both photonic effects need to be taken into account to optimize photonic structure design for a specific application.

**Comparison of simulation and experiment.** We compare simulation and experiment by varying the design wavelength $\lambda_{design}$, the excitation wavelength $\lambda_{excitation}$, and the irradiance. Thereby, we performed the measurements at an irradiance around ~500 suns in order to gain a good signal-to-noise ratio in all parameter scans, which was not feasible at only 1 sun illumination. We use 40 fabricated sample designs around the maximum UC enhancement expected from theory to investigate the dependence of UCPL$_{rel}$ on $\lambda_{design}$ (Fig. 4a). For evaluation, we sort the data into five groups (I–V) of similar $\lambda_{design}$. Both, the active- and TiO$_2$ layer are scaled to match the desired design wavelength $\lambda_{design}$. The corresponding active layer thickness is shown in the

top $x$-axis of Fig. 4a. The simulated UCPL$_{rel}$ is the same as in Fig. 3e, including the standard deviation of the layer thicknesses and both photonic effects. In experiment, the photonic effects increase the UC signal for $\lambda_{design}$ around the simulated maximum UCPL$_{rel}$ at 1855 nm. In groups II and III, at and close to the simulated maximum, respectively, the highest mean measured UCPL$_{rel}$ is found, while group III slightly outperforms group II. Moving further away from the maximum, the experimentally measured UCPL$_{rel}$ is lower (groups I and IV), and finally suppressed when the $\lambda_{excitation}$ falls into the photonic bandgap (group V). We expect that the main reason for the variation of the single UCPL$_{rel}$ measurements, also within the same design wavelength, are slight thickness variations of the single layers in each stack that appear due to production inaccuracies ("Methods"). Despite these thickness variations of single layers, a defined design wavelength can be assigned to each sample we investigated (Supplementary Fig. 7). In simulation, we also take the impact of the production inaccuracy on UCPL$_{rel}$ into account. However, the simulation features the mean expected reduction over 1000 separate calculations. Most random thickness variations of single layers lead to a decrease in energy density in the active layers and therefore to a reduced UCPL$_{rel}$. For particular designs though, non-periodic thickness variations of single layers can lead to an additional strong increase of the energy density in the active layers[55], which consequently leads to an additional increase in UCPL$_{rel}$. This might contribute to a maximum measured enhancement of 4.1 for $\lambda_{design} = 1844$ nm. A closer analysis of the impact of the non-periodicity within each single Bragg structure design is out of scope of this paper.

Next, the effects of varying $\lambda_{excitation}$ are important for applications with a broad-band excitation source, such as photovoltaics. We use the same groups as in Fig. 4a, and evaluate the mean and standard deviation of UCPL$_{rel}$ within each group ("Methods") (Fig. 4b (i–v)). In simulation, $\lambda_{excitation}$ is varied over the complete absorption range of the upconverter material Er$^{3+}$ (Supplementary Fig. 1), featuring the center $\lambda_{design}$ of each group.

In experiment, we covered the range of $\lambda_{\text{excitation}}$ between 1500 nm and 1560 nm. Group II shows a broad plateau for $\lambda_{\text{excitation}}$ around 1523 nm. This corresponds to the expectation that for $\lambda_{\text{design}} = 1855$ nm, UCPL$_{\text{rel}}$ peaks at $\lambda_{\text{excitation}} = 1523$ nm. For group I, the maximum enhancement is expected at a shorter $\lambda_{\text{excitation}} = 1465$ nm, for groups III, IV, and V at longer $\lambda_{\text{excitation}}$ of 1555 nm, 1585 nm, and 1645 nm, respectively. Consequently, in the investigated $\lambda_{\text{excitation}}$ range, the dependence of UCPL$_{\text{rel}}$ on $\lambda_{\text{excitation}}$ corresponds to a falling flank (group I), a rising flank (groups III and IV), or a rather flat region (group V). The slope expected from simulation, which characterizes the Bragg structures effects, is very well visible in the experimental data in all five groups. We performed the same evaluation for the UC emission around 814 nm (Supplementary Fig. 12) and found the same good agreement between simulation and experiment. With a suitable design for a specific application, UCPL$_{\text{rel}}$ can be increased in any desired spectral region. We find that in the optimum design range (group II), the complete core domain of the Er$^{3+}$ absorption spectrum between about 1475 nm to 1575 nm can be significantly enhanced, with a simulated peak UCPL$_{\text{rel}}$ of 2.4 at an irradiance of 1.48 W cm$^{-2}$. At the outer ranges of the absorption domain (visible particularly in groups I and IV), the enhancement factors are slightly higher. This is because in spectral regions where very little light is absorbed, the photonic enhancement has a larger impact on UC efficiency than in spectral regions with higher absorptance. In the mean of all 2440 separate parameter combinations in the excitation wavelength scan, we find that the mean agreement of measurement and simulation, the measured UCPL$_{\text{rel}}$ divided by simulation, lies at $81.8 \pm 23.9\%$ ("Methods"). For such a large number of measurements, one could expect, that the mean of experiment and simulation should match, especially because we already take into account reductions of UC enhancement due to production inaccuracies. We expect that there are two reasons for this additional reduction of UCPL$_{\text{rel}}$ that we see in the mean of all measurements: (i) the distribution of upconverter nanoparticles within the active layers, and (ii) the surface roughnesses in the Bragg structure. The photonic effects are strongest in the center of the active layer. However, the upconverter nanoparticles are not evenly distributed in the active layer, they are rather positioned at the outer ranges (compare to Fig. 1b). Additionally, the layers of the Bragg structure feature a roughness of around 10 nm (Supplementary Fig. 8b), which introduces additional scattering that most probably leads to a reduction of the overall photonic effects on UCPL$_{\text{rel}}$, which is currently not accounted for in the model.

Finally, in Fig. 4c, we demonstrate the dependence of the photonic effects on the irradiance for a sample of group III with $\lambda_{\text{design}} = 1888$ nm. The simulation is again plotted with only the effect of the energy density $u_{\text{rel}}$ taken into account, only the LDOS effect, and for both effects. Considering only the effect of the energy density results in a falling curve for UCPL$_{\text{rel}}$ toward higher irradiances. In the low irradiance regime, in which the reference performs poorly, an increase in energy density, followed by a stronger absorption, largely increases the probability of an energy transfer UC process to take place, resulting in a high UCPL$_{\text{rel}}$. This becomes evident when looking at the absolute UCPL simulated down to 1 sun irradiance (Supplementary Fig. 11a). Consequently, also the UC quantum yield increases significantly at low irradiances (Supplementary Fig. 11b and c). At higher irradiances, energy transfer UC to yet higher energy levels becomes more probable, which decreases the probability of our main UC emission at 984 nm, thus decreasing UCPL$_{\text{rel}}$. In direct comparison with experiment, one can see that the absolute value of UCPL$_{\text{rel}}$ is reproduced, but that the effect of a falling UCPL$_{\text{rel}}$

toward higher irradiances is exaggerated. However, the negative effect of the LDOS is stronger in the low irradiance regime, as can be seen from the curve showing only the LDOS effect. Thus, when both effects are taken into account, the experimental data clearly follows the slope of the simulation, accurately reproducing the simulated UCPL$_{\text{rel}}$. The mean of all 41 measurements contained in the irradiance scan lies at $104.5 \pm 11.6\%$ of the simulated UCPL$_{\text{rel}}$.

For all separate 2480 parameter combinations in the excitation wavelength and irradiance scan, the mean UCPL$_{\text{rel}}$ in experiment lies at $82.2 \pm 24.0\%$ of the simulation ("Methods").

## Discussion

The most important aspect of this work is the exact experimental validation of a comprehensive simulation modeling framework, describing 1D-photonic structure effects on photon UC in embedded upconverter nanoparticles. The experimentally measured UC enhancement precisely features the expected values and behavior from simulation in all three performed parameter scans reaching $82 \pm 24\%$ of the simulated UC enhancement in the mean of all 2480 measurements with different parameter combinations. Taking into account the complexity of both simulation model and experiment, with all three involved parameter scans, the agreement within one standard deviation is very good. We demonstrated that it is of crucial importance to include both photonic effects of a varied local energy density and modified LDOS, as well as internal UC dynamics and production inaccuracies to optimize a photonic structure design. The principle of UC enhancement due to photonic effects can be applied to any kind of upconverter material with a similar set of energy levels as Er$^{3+}$. An inaccuracy that the model currently features is a slight overestimation of the LDOS effect because we simulate it for an infinite photonic crystal. However, this method allows for an investigation of directionality of UC emission, implemented as a fractional LDOS[56], which will be subject of our future work.

We chose to investigate a simple Bragg structure with only four active layers to be able to tune and understand all appearing effects. The production accuracy we reached in the experiments allowed for a detection of all the expected photonic effects. Even though Bragg structures might not be the photonic structures showing the highest UC enhancement factors, they have many features that are important and promising for UC enhancement for an application in photovoltaics: The amount of upconverter material, and thereby the overall absorption, is not limited by the design but can be adapted by adding more layers to the stack. Furthermore, as we could show, UCPL enhancement occurs in a broad spectral range, covering most of the investigated Er$^{3+}$ absorption range. We additionally investigated the relative UCPL under a varied incident angle both theoretically and experimentally. The analysis is shown in Supplementary Fig. 13, where the very good agreement of theory and experiment is documented. We find that light can be efficiently coupled into the structure up to large incident angles of about 30°. This is important for broadband, wide-angle applications like photovoltaics: for a simple system without tracking, the movement of the sun means varying incident angles, but also for concentrator systems using tracking, the concentration means that the angular range of the light incident onto the solar cell is increased. In conclusion, our analysis showed that a Bragg structure has spectral and angular characteristics that are beneficial for the application in photovoltaics.

Drawing a thorough comparison to literature is difficult due to the difference in photonic structure design and choice of reference, quality of upconverter material, and measurement setup, such as the applied irradiance and detection angle, which all

greatly influence the resulting UCPL enhancement and are often not fully reported. However, with only four active layers, the maximum UCPL enhancement factor of 4.1 at 1.48 W cm$^{-2}$ irradiance is a good achievement, especially when taking into account that we detected the UCPL signal integrated over all angles. In the Supplementary Tables 1–4, we provide an overview of design and experimental details for the photonic upconverter devices from literature that we discussed in the introduction. In most literature, the measured maximum enhancement at one distinct angle is reported. Lin et al.[43] report a giant UCPL enhancement of $10^4$ in a waveguide structure[43]. However, this enhancement only occurs in an excitation angle range of ~1° and off the optimum angle it drops rapidly by three orders of magnitude. This is not favorable for broad-band applications with varying excitation angle, such as photovoltaics. Johnson et al. also investigated a Bragg structure of $Er^{3+}$-doped porous silicon. Under 1550 nm excitation and a high irradiance, they report a 26.5- and 5-fold enhancement of the green and 980 nm UC emission, respectively, for a structure similar to what we define a 30 active layer structure[46] (see Supplementary Table 4). The enhancement occurs in an incident angle range of ~4°. They mention difficulties in controlling the layer thickness, which crucially diminishes the photonic effects. This report agrees well with our simulation, including the correct refractive indices and a large layer thickness variation (discussed in ref. [50]) and pronounces the importance of including fabrication inaccuracy: a precise 4-active layer stack can reach an effect close to an imprecise 30-active layer stack. The amount of upconverter material, of course, also needs to be considered: while the design used by Johnson et al. features a total thickness of all upconverter-doped layers of as much as ~15 µm, our investigated Bragg structures with four active layers features a summed up active layer thickness of ~1.3 µm. Rojas-Hernandez et al. report a 25-fold enhancement of green UCPL under 975 nm excitation in a microcavity structure of 21 layers of $TiO_2$ and $Tb^{3+}/Yb^{3+}$-doped aluminosilicate glass, featuring a summed up active layer thickness of ~1 µm, measured at a distinct detection angle[44] (see Supplementary Table 3). In comparison, for a Bragg structure with ten active layers (in total 21 layers) in our current production accuracy, from simulation we expect a UCPL enhancement of a factor of 4.3 at a relatively high irradiance of 1.48 W cm$^{-2}$ and 27 at a low irradiance of 1 sun.

In summary, with our comprehensive simulation model that we experimentally validated in this work, 1D-photonic crystals can now be thoroughly optimized for specific applications. We identify Bragg structures as promising and flexible for UC enhancement for broad-band applications that feature incident angle variations, such as photovoltaics.

There are several ways to further improve the efficiency of such photonic upconverter devices: (i) increasing the number of layers in the Bragg structure largely increases the UCPL enhancement, e.g. from a factor of 4.4 at 1-sun irradiance for the layer stack we investigate in this manuscript with four active layers, to a factor of 66.6 for a layer stack with 20 active layers, considering the same production accuracy. (ii) Using a material with a higher refractive index for the high-refractive index layer of the Bragg structure also increases the photonic effects[57]. (iii) Applying down-shifting materials for spectral concentration into the absorption range of the upconverter could increase the used spectral fraction as well as the irradiance acting on the upconverter and therefore increase the efficiency at lower irradiance levels[58–60]. Superior UC properties have already been demonstrated for hybrid upconverter materials of lanthanide-doped upconverter nanoparticles combined with organic dyes as sensitizers, and have been applied to photovoltaic systems[13]. Down-shifting the complete spectral range below the bandgap of Silicon into the absorption range of

the upconverter Erbium, is estimated to increase the current enhancement in the silicon solar cell by a factor of three in comparison to a purely Erbium-based system[61]. (iv) Via rear-side mirrors, the irradiance in the photonic upconverter device could be further increased. (v) In addition, concentration optics would allow to operate in an irradiance regime in which the upconverter features a higher UC quantum yield, such as in conventional concentrator modules[54] or in devices with concentrator optics specifically designed for UC[62]. With these measures combined, an optimistic estimate is to generate an additional current of 1.7 mA cm$^{-2}$ in a silicon solar cell[61]. Especially in silicon-based tandem solar cells, in a situation where the silicon bottom-cell is limiting the overall current, this could have a significant impact on overall performance. To reach this goal, further progress and optimization in all mentioned areas is necessary. The contribution of this paper is to provide a validated theoretical model to enable a knowledge-based optimization process of photonic upconverter devices.

## Methods

**Optimization of active layers.** The low refractive index layers of the Bragg structure are composed of PMMA (120,000 g mol$^{-1}$, Sigma-Aldrich), containing 25 wt% of upconverter nanoparticles. These layers are referred to as active layers. The core-shell upconverting nanoparticles are made of hexagonal sodium yttrium tetrafluoride ($\beta$-NaYF$_4$) with a 25% doping of trivalent erbium ($Er^{3+}$) ($\beta$-NaYF$_4$:25% $Er^{3+}$) and an inert $\beta$-NaLuF$_4$ shell, produced as reported in ref. [63] with oleic acid ligands. We produced thin active layers via spin-coating from a mixture of upconverter nanoparticles and 4 wt% or 5 wt% PMMA in toluene. We performed the spin-coating process (Specialty Coating Systems G3P-8) for 60 s with 250 µL of solution. Active layers on top of thin $TiO_2$ layers we produced with different spin-speeds between 500 r.p.m. and 2000 r.p.m. and measured the resulting thickness with an atomic force microscope (AFM, Dimension Edge, Bruker). The relation between spin-speed and thickness was drawn from a fit to the data with an empiric model, which then allowed the precise production of the desired layer thickness, with a production accuracy of 1.3%, corresponding to 4.15 nm. We determined the refractive index of the active layers via spectroscopic ellipsometry (M-2000, J.A.Woollam Co., USA). For data analysis, we applied a Cauchy model, implemented in the Complete Ease[64] software, yielding a refractive index of the active layer of 1.474 at 1523 nm wavelength. More detailed information on optimization of the active layers can be found in the Supplementary Note 1.

**Optimization of $TiO_2$ layers.** The high refractive index layers of the Bragg structure are made of $TiO_2$, which we deposited via atomic layer deposition (ALD) (R-200 Advanced, Picosun, Finland). The ALD process was run at a chamber temperature of 100 °C from molecular precursors $H_2O$ and $TiCl_4$ (purchased from Sigma-Aldrich (≥99% $TiCl_4$)). Via X-ray diffraction (XRD) measurements (XRD D8, Bruker), we confirmed that the produced $TiO_2$ films are amorphous. We analyzed the layer thickness and refractive index of $TiO_2$ layers via spectroscopic ellipsometry, as described above, but in this case utilizing a Cody Lorentz model. At a wavelength of 1523 nm, the determined refractive index lies at 2.279. We adapted the layer thickness by varying the number of deposition cycles. From thickness measurements of single layers, we determined the production accuracy, featuring a standard deviation of the mean of 0.8%, corresponding to 1.53 nm. This value served as input parameter in the simulation of non-ideal Bragg structures. More detailed information on optimization of $TiO_2$ layers can be found in the Supplementary Note 2.

**Production of optimized Bragg structures and reference samples.** We fabricated optimized Bragg structures out of five $TiO_2$ layers and four intermediate active layers. For production, we alternately carried out the processes of atomic layer deposition for $TiO_2$ and spin-coating for active layers in one glovebox in Argon atmosphere to reduce contamination of the samples. The simulated maximum UCPL enhancement, due to the photonic effects of the Bragg structure, appears at 1855 nm design wavelength. This corresponds to a layer thickness of 203 nm for $TiO_2$ and 315 nm for the active layers. We fabricated eight different samples with target design wavelengths right at, as well as longer and shorter than, the expected maximum enhancement. Each sample was placed at a distinct position in the ALD chamber and for each precisely determined layer thickness of $TiO_2$, we spin-coated the matching active layer thickness to gain the same optical thickness of both layers and therefore a defined design wavelength.

As a reference, we choose a stack of only the active layers of the corresponding Bragg structure. This way, the reference contains the same amount of upconverter material without the photonic structure around it. We fabricated the reference samples by spin-coating the active layers of the corresponding Bragg structure right on top of each other. Thereby, the target thickness of the active layers in Bragg

structure and reference is identical. The spin-coating parameters were adapted for each substrate material separately (Supplementary Fig. 2b).

**Design characterization of Bragg structures.** On each sample, we characterized five distinct points. With a spectrophotometer (Lambda 950, PerkinElmer, Germany), we measured the characteristic Bragg structure reflectance for each sample point at a tilt of 8° relative to the incident beam. Using an aperture, the beam diameter was reduced to ~1 mm diameter. We performed the simulation of the Bragg structure reflectance in an implementation of the transfer matrix method[50], subsequently comparing each measured reflectance to the simulated reflectance, scanning through design wavelengths and calculating the squared difference. For each sample point, the design wavelength for which the simulation features the minimum squared difference to the measured curve, explicitly determines its design (Supplementary Fig. 7).

**UCPL measurement setup.** A sketch of the UCPL measurement setup can be found in Supplementary Fig. 9. A tunable low power 20 mW infrared laser (TSL-510, Santec) served as excitation source. We measured all samples in an integrating sphere (819C-SL-5.3, Newport), placed in a center mount holder with a tilt of 4° relative to the incident laser beam. A 75 mm focal length lens was additionally installed at the entrance port of the integrating sphere to avoid unwanted coherence effects in the glass substrate. The signal was detected with a spectrograph (SP2300i, Princeton Instruments, USA), equipped with a blazed grating (150 grooves mm$^{-1}$ at a blaze wavelength of 800 nm) and a silicon CCD detector (PIXIS:256E, Princeton Instruments, USA).

To extract the real emitted spectrum of a measured sample, we corrected the signal for the spectral response of setup components like grating, detector, and lens. A calibrated tungsten halogen lamp served as excitation source for measuring the spectral response correction function of the setup.

For all laser powers and excitation wavelengths used in the experiments of this work, we determined the irradiance of the excitation beam at the sample position. We measured the area of the laser beam with a beam profiler (BP209-IR/M, Thorlabs) and determined the laser power with a photodiode sensor (PD300-IR, Ophir Photonics). Because the UC process is non-linearly dependent on the irradiance, we choose to calculate the laser irradiance only from the FWHM region of the Gaussian-shaped laser profile. This way, the high irradiance region within the laser profile, which is more relevant for the UC process, is calculated more precisely. Additionally, we scaled the laser area with a factor of 1/cos(4°) to account for the tilted sample (Supplementary Fig. 10).

**UCPL measurements.** We analyzed the UCPL at the same five points on each sample that were characterized in spectrophotometer measurements (Supplementary Fig. 7). For both, design wavelength scan (Fig. 4a) and irradiance scan (Fig. 4c), we measured UCPL spectra with 200 s integration time, for the excitation wavelength scan (Fig. 4b) with 60 s integration time. We calculated the UCPL$_{rel}$ as the ratio of integrals over the UCPL spectra of Bragg structure and reference, within the wavelength range of 930–1020 nm (compare to Fig. 2c).

**Calculation of mean agreement of measurement and simulation.** We choose to quantify the mean agreement of measured and simulated UCPL$_{rel}$ for all 2480 measurements with separate parameter combinations. The measured UCPL$_{rel, measured}$ is compared to the exact same parameters in simulation UCPL$_{rel, simulated}$, with a binning of 1 nm in design wavelength and featuring the exact irradiance of experiment. We then calculate the mean and standard deviation of UCPL$_{rel, measured, i}$ divided by UCPL$_{rel, simulated, i}$ for all measurements $i$ within the evaluated group of measurements.

For the excitation wavelength scan that we visualize in Fig. 4b, we use the simulation at each excitation wavelength in steps of 1 nm, each featuring the exact irradiance of experiment (Supplementary Fig. 10). For the final quantification, yielding 82 ± 24%, we include all measurements with different parameter combinations. This includes all measurements of the excitation wavelength scan (Fig. 4b) and all (except one) measurements of the irradiance scan (Fig. 4c). We do not include the design wavelength scan (Fig. 4a), as these measurements are a repetition of the measurements in the excitation wavelength scan at 1523 nm, as well as the measurement at 1.48 W cm$^{-2}$ in the irradiance scan, also being a repetition.

**Simulation of local energy density.** We here give a brief overview of the comprehensive simulation model; a detailed description of the model as well as the applied simulation details can be found in ref. [50].

We calculate the local energy density $u(x)$ within the Bragg structure and reference using an implementation of the transfer matrix method[50]. The relative local energy density $u_{rel}(x)$ of the Bragg structure is calculated relative to the reference as a half-infinite low refractive index material. For visualization, we also define the mean relative energy density as the quotient of the integral over $u(x)$ of the Bragg structure only within the active layers and the reference. These quantities are very sensitive to structural imperfections[50]. Therefore, we perform the simulation as close to the experiment as possible. Via Monte-Carlo simulations, we

modify the thickness $d$ of each layer of the Bragg structure as $d \rightarrow d + \delta d$, whereby the $\delta d$ is a random value of a Gaussian distribution with standard deviation $\sigma$. In this work, we calculate the average $u(x)$ over 1000 separate calculations. The experimentally measured standard deviations of the single layer production accuracies of 4.2 nm and 1.5 nm for the active and TiO$_2$ layers, respectively, serve as input parameters. $u_{rel}(x)$ is the exact calculation of the energy density at each position in the Bragg structure, however, it is difficult to visualize this value in dependence on a varied Bragg structure design. Therefore, we need an average value in only the active layers of the Bragg structure, to be able to easily visualize the dependence of $u_{rel}(x)$ on a varied design wavelength. The average relative energy density $\bar{u}_{rel}$ (Fig. 3b) we thus define as the integrated $u(x)$ only in the active layers of the Bragg structure, divided by the integral over $u(x)$ in the reference.

**Simulation of LDOS.** The LDOS for infinite photonic crystals can be derived from Eigenmode calculations. We use the software package MIT Photonic Bands[65] and subsequently the histogramming method[56] to calculate the 3D LDOS. This dimensionless LDOS can be mapped to any unit cell size $a$, given by the design wavelength and considered transition frequency, given by the transition wavelength $\lambda_{fi}$ from an initial state $i$ to a final state $f$. We calculate the modification of the LDOS due to the Bragg structure relative to the homogeneous reference, consistent of only the low refractive index material, yielding $\text{LDOS}_{rel}\left(x, \omega'_{fi}\right)$. We choose this approach because it allows for analyzing the LDOS for different emission angles, which is subject of our future work. However, the fact that the calculation is done for an infinite photonic crystal overestimates the effect of the LDOS for the structure we are analyzing in this work with only nine layers in total. Again, as described above, for visualization, we also define the average relative LDOS $\overline{\text{LDOS}}_{rel}$ (Fig. 3a), as the integral over $\text{LDOS}\left(x, \omega'_{fi}\right)$ in only the active layers of the Bragg structure divided by the integrated $\text{LDOS}\left(x, \omega'_{fi}\right)$ in the reference and scaled to the regarded emission frequency.

**Rate equation model.** We describe the dynamics of the UC process in a rate equation modeling framework, developed in ref. [52] for homogeneous media. Based on coupling plasmonic effects with the rate equation model[66,67], we extended the model for a photonic environment in ref. [51] and ref. [68]. The determination of experimental input parameters on UC dynamics are described in ref. [69]. The model version used in this work is published in ref. [50].

Compared to the current model version, the simulation methods used in Hofmann et al. 2016 (refs. [68]) were slightly different, as pointed out in the publication of the more advanced version in Hofmann et al. 2018 (ref. [50]). Two changes are significant: (i) in Hofmann et al. 2016, only ideal Bragg structures were investigated, no production accuracies are included. (ii) Furthermore, there has been a small bug in the simulation script of the LDOS in Hofmann et al. 2016, which has an impact on the trends visible in the graphs including the effect of the LDOS. This bug was fixed in the version published in Hofmann et al. 2018 and we are currently working on an Erratum to the paper Hofmann et al. 2016 to correct the errors. The main findings that are discussed in Hofmann et al. 2016, however, are not influenced by this error and will not change in the Erratum.

The rate equation model describes the population of the Er$^{3+}$ energy levels in the host crystal $\beta$-NaYF$_4$ (Fig. 2a). The rate of change of the occupation density vector is described by

$$\dot{\mathbf{n}} = [M_{GSA} + M_{ESA} + M_{STE} + M_{SPE} + M_{MPR}] \cdot \mathbf{n} + \mathbf{v}_{ETU}(\mathbf{n}) + \mathbf{v}_{CR}(\mathbf{n}), \quad (1)$$

taking into account the transition matrices $M$ describing the probabilities of the linear processes ground-state absorption (GSA), excited-state absorption (ESA), stimulated emission (STE), spontaneous emission (SPE), and multi-phonon relaxation (MPR), as well as the non-linear Förster energy transfer processes, energy transfer UC (ETU), and cross relaxation (CR). The energy density influences all stimulated processes. This change is accounted for by multiplying the probabilities of all absorption and STE processes with the relative local change in energy density:

$$M_{GSA} \rightarrow M_{GSA}\, u_{rel}\left(x, \omega'_{fi}\right),\; M_{ESA} \rightarrow M_{ESA}\, u_{rel}\left(x, \omega'_{fi}\right),\; M_{STE} \rightarrow M_{STE}\, u_{rel}(x, \omega'_{fi}).$$
$$(2)$$

According to Fermi's golden rule, a modified LDOS influences the probability of SPE processes. The Einstein coefficients $A_{fi}$, describing SPE within the matrix $M_{SPE}$, are therefore multiplied with the relative local change of the LDOS:

$$A_{fi} \rightarrow A_{fi}\, \text{LDOS}_{rel}\left(x, \omega'_{fi}\right). \quad (3)$$

The UCPL from energy level $^4I_{11/2}$ to $^4I_{15/2}$ is the main focus of this work. UCPL for one emission is calculated from the probability of the emission, given by the Einstein coefficient and the current population of the initial energy level $N_i$. For the Bragg structure, we perform the calculation at all positions of the upconverter material, so across all active layers. The relative UCPL of Bragg structure (brg) and

homogeneous reference (ref) is given by

$$\text{UCPL}_{\text{rel}} = \frac{\int A_{fi,\text{brg}}(x) N_{i,\text{brg}}(x) dx}{A_{fi,\text{ref}} N_{i,\text{ref}} \times x}. \qquad (4)$$

Absorption of the upconverter material is included in the REM as the relative absorption spectrum of $\beta$-NaYF$_4$:Er$^{3+}$ (Supplementary Fig. 1).

## Data availability

Data that support the findings of this study are available from the corresponding author upon reasonable request. Source data are provided with this paper.

## Code availability

The code used for all simulations in this manuscript is freely available for download at https://doi.org/10.24406/fordatis/110.2.

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

## Acknowledgements

The authors would like to thank Oliver Höhn and Dmitry Busko for fruitful discussions; Felicia Volle, Volker Kübler, Kristina Winkler, Fabian Gerspacher, Nicolo Baroni, and Felix Martin for measurements. The research leading to these results has received funding from the Baden-Württemberg Ministry of Science, Research and Arts, the Baden-Württemberg Ministry of Finance and Economy as well as from Fraunhofer-Gesellschaft in Munich in the project "NaLuWiLeS: Nano-Strukturen zur Lumineszenzverstärkung für die Wirkungsgradsteigerung von LEDs und Solarzellen" by the Sustainability Center Freiburg. The atomic layer deposition process was carried out at the Karlsruhe Nano Micro Facility (KNMF, www.knmf.kit.edu), a Helmholtz Research Infrastructure at Karlsruhe Institute of Technology. C.L.M. Hofmann gratefully acknowledges the scholarship support from the Heinrich-Böll Stiftung, and S. Fischer gratefully acknowledges the scholarship support from the German Research Foundation (DFG, agreement FI 2042/1-1). E.H. Eriksen gratefully acknowledges the support from Innovation Fund Denmark through the SunTune project. B.S. Richards gratefully acknowledges the following funding from the Helmholtz Association: (i) professorial recruitment initiative; (ii) the Helmholtz Energy Materials Foundry (HEMF); (iii) the Science & Technology of Nanosystems (STN) research program.

## Author contributions

C.L.M.H. performed the simulations, sample design and preparation, experiments and data analysis, and wrote the manuscript. S.F. produced the upconverter nanoparticles. E.H.E. adapted and tested the simulation code. B.B. gave significant advice to the photoluminescence setup and upconversion photoluminescence data interpretation. C.R. adapted and was in charge of the atomic layer deposition process for sample production. D.Y. constructed the measurement setup for upconversion photoluminescence measurements under a varied incident angle and performed the measurements at this setup. I.A.H. gave significant advice on calibration, planning, and interpretation of photoluminescence measurements. B.S.R. significantly contributed to planning the experiments, gave significant advice on ellipsometric data analysis and upconversion photoluminescence data interpretation, and conceived of the TiO$_2$ thin-film deposition process for multilayer-stack production. J.C.G. led the work and conceived of the research question, contributed significantly to planning the experiments, as well as to the interpretation of experimental and simulation data. All authors discussed the results and contributed to the finalization of the manuscript.

## Funding

## Competing interests

The authors declare no competing interests.
