## [Peer Review File · Nature Communications]

Reviewers' comments:

Reviewer #1 (Remarks to the Author):

In this manuscript, the authors investigate upconversion processes in 1D photonic crystals. They begin by modeling and measuring the properties of the cold cavity consisting of Bragg mirrors, and show good agreement with experiment. Then they measure upconversion of rare earth-doped core-shell nanoparticles with and without the cavity, and observe substantial enhancement (approximately 4x) with an input power of 14.8 suns. In their model, they show that this type of enhancement depends on the local density of states and the average relative energy density. They subsequently are able to show that the upconversion enhancement spectrum and power dependence scales similarly to the model presented, thus providing initial confirmation for the accuracy of their model within the conditions studied.

Overall, this work is of interest, but also brings up the following questions, which should be addressed before considering this work further for publication in Nature Communications:

1. Many studies looking at upconversion, including this one, focus on intensities well above one sun. In this manuscript, what is the reason for choosing an irradiance of 1.48 W/cm²?
2. Also, what assumptions are made in equating the irradiance of 1.48 W/cm² with 500 suns? If one were to just divide by the solar constant, one would obtain 14.8 suns, but the spectra would also be much different.
3. Previous studies have also observed non-linear scaling of higher absolute output powers with higher input powers, akin to a harmonic generation process. When upconversion is presented as a relative factor, as in this manuscript, it is not necessarily as obvious that this remains the case. Can the authors show how the absolute upconversion power should scale down from the intensity they measured down to an intensity of one sun, based on their model?
4. If these values are much smaller than the efficiency of direct solar conversion, can the authors briefly comment on what additional improvements or modifications would be needed to make these relevant to improving champion cell efficiencies?
5. If we compare to the modeling framework developed by the authors in prior work (e.g., C.L.M. Hofmann et al., Opt. Express 26, 7537, 2018 and C.L.M. Hofmann et al., Opt. Express 24, 14895, 2016), are there any significant changes? Section 3.10 seems to indicate that there are not many, but the upconversion graphs and trends (e.g., Figs. 2b, 3c, and 4) look significantly different.
6. In Fig. 4a, the upconversion data has a fair spread. What is the main source of this spreading in the vertical direction at each design length?
7. Also, given the current data in Fig. 4a, how confident can one be about the simulated peak conversion taking place with a design wavelength around 1850 nm (and an active layer thickness around 630 nm)?

8. In the discussion, since some of the prior results take place over a narrow angular range, is it possible to estimate the photovoltaically-relevant enhancement for each system (including this one) at equal intensities? Then you would have the option of presenting a table summarizing the upconversion efficiency associated with each experimental photonic upconversion structure. That would make it much more clear whether this particular 1D photonic crystal has any special properties in its own right, or if the advance is primarily in validating the model.

If the authors are able and willing to address these comments, I will be available to review their responses again, if that would be helpful. Thank you.

Reviewer #2 (Remarks to the Author):

In this manuscript, the authors investigate the effects of 1D-photonic structure on photon UC in embedded upconverter nanoparticles through both simulation and experimentation. Their results show the experimental validation of a comprehensive simulation modeling framework. By studying the parameters of active layer thickness, irradiance, and excitation wavelength, the authors prove that experimentally measured UC enhancement features the expected values and behavior from the simulation in all three performed parameter scans, reaching $82\pm 24\%$ of the simulated UC enhancement out of 2480 measurements with different parameter combinations. In fact, in the past decades, the effect of photonic crystals on the luminescence of embedded light-emitters has been extensively studied, both experimentally and theoretically. For example, GdVO₄:Eu³⁺ nanophosphors were sandwiched between two photonic multilayers based on transparent ZrO₂ and SiO₂ layers to increase the needed emission (Nanoscale Horizon, DOI: 10.1039/c8mh00123e). Many works have utilized photonic crystals to extract the emission of LED. I feel this work lacks the novelty for Nature Comm. A few other comments for improvement:

1. For 1D photonic structure, reflection spectrum is dependent on the incident angle. In this work, what angle was the reflection spectrum measured?
2. The upconversion photoluminescence is measured in an integrating sphere. Due to the inherent directional nature of photonic crystals, the enhancement effect should also be directional. The authors should compare the integrated effect and one-direction effect by simulation and experiments.
3. It is well known the PMMA has an absorption in the NIR region. This effect needs to be taken into consideration.
4. The thicknesses of high and low refractive index materials determine the bandgap of photonic crystals. This work only investigated the effect of active layer thickness. The thickness of TiO₂ should also be considered.

Reviewer #3 (Remarks to the Author):

The manuscript by Hofmann and co-workers reports a very careful study on the effect of Bragg structures on rare-earth upconversion. They show that their experimental realisations come close to theoretical expectations, validating both.

Upconversion is an area of intense interest, and the recent review by Jin and others in Nature Photonics exemplifies this. The photonic approach to upconversion enhancement is one that demands careful reports of the kind presented here.

The paper is well written, and the reference list appears appropriate, with the exception perhaps of the above review.

If I were to find one criticism, it is that I do not find the absolute upconversion efficiencies which satisfy the reader that progress is being made in the field. Several of the authors are thought leaders in this regard, so it is surprising that it is missing.

Otherwise, I would recommend publication as is, subject to minor language tweaks by the editorial process.

Many thanks to all reviewers for their detailed and thoughtful comments and questions. They were very helpful for us to clarify and improve our manuscript. We particularly worked on clarifying the motivation of different aspects of our research and included new comprehensive passages in the manuscript to explain the choice of investigated structures and application relevance. Supporting the question of application relevance, we also added two new sections to the Supplementary information on i) simulations of upconversion efficiency at solar irradiance and ii) a theoretical and experimental analysis of the photonic structure effects under a varied incident angle. Please find our detailed point-by-point response to all comments and concerns raised below.

The *reviewer's comments and questions* are marked as italic text. Passages we **deleted** in the manuscript are highlighted in red, **new** passages we took in are highlighted in blue. Please also find a new uploaded version of the manuscript and supporting information with highlighted changes, as well as the source data file.

Reviewer #1 (Remarks to the Author):

In this manuscript, the authors investigate upconversion processes in 1D photonic crystals. They begin by modeling and measuring the properties of the cold cavity consisting of Bragg mirrors, and show good agreement with experiment. Then they measure upconversion of rare earth-doped core-shell nanoparticles with and without the cavity, and observe substantial enhancement (approximately 4x) with an input power of 14.8 suns. In their model, they show that this type of enhancement depends on the local density of states and the average relative energy density. They subsequently are able to show that the upconversion enhancement spectrum and power dependence scales similarly to the model presented, thus providing initial confirmation for the accuracy of their model within the conditions studied.

Overall, this work is of interest, but also brings up the following questions, which should be addressed before considering this work further for publication in Nature Communications:

1. Many studies looking at upconversion, including this one, focus on intensities well above one sun. In this manuscript, what is the reason for choosing an irradiance of 1.48 W/cm²?

We chose to use an irradiance of 1.48 W cm⁻² in experiments, which corresponds to ~500 suns concentration, because this is a regime typical for high-concentration PV systems¹. Thus, a certain relevance is given for photovoltaics as our target application, unlike in other experimental studies in literature where much higher (and hence unrealistic) irradiances are used. Working at only one sun was not possible because here it would have been difficult to achieve a good signal to noise ratio in our measurements for all excitation wavelengths we scanned, which was necessary for the confirmation of our model.

We deleted the shortened explanation from line 182:

"..in the absorption range of the upconverter Er³⁺ from 1450 nm to 1600 nm⁴⁵."

And clarified our approach by adding the following passage in the manuscript in line 144 – 148:

"In Fig. 3, we investigate the change in UC emission due to photonic effects at two different irradiance levels that are relevant for the target application of photovoltaics: i) at one sun illumination, where the irradiance in the absorption range of the upconverter Er³⁺ between 1450 nm to 1600 nm is 3 mW cm⁻²⁵¹ and ii) at 1.48 W cm⁻², corresponding to ~500 suns concentration, which is a typical regime for high-concentration photovoltaic systems⁵²."

And further added in line 210 – 212:

"Thereby, we performed the measurements at an irradiance around ~500 suns in order to gain a good signal-to-noise ratio in all parameter scans, which was not feasible at only one sun illumination."

51. Fischer, S., Fröhlich, B., Steinkemper, H., Krämer, K. W. & Goldschmidt, J. C. Absolute upconversion quantum yield of β -NaYF₄ doped with Er³⁺ and external quantum efficiency of upconverter solar cell devices under broad-band excitation considering spectral mismatch corrections. *Sol. Energy Mater. Sol. Cells* **122**, 197–207 (2014).
52. Wiesenfarth, M., Anton, I. & Bett, A. W. Challenges in the design of concentrator photovoltaic (CPV) modules to achieve highest efficiencies. *Appl. Phys. Rev.* **5**, 41601 (2018).

2. Also, what assumptions are made in equating the irradiance of 1.48 W/cm² with 500 suns? If one were to just divide by the solar constant, one would obtain 14.8 suns, but the spectra would also be much different.

We consider the irradiance only in the absorption range of the upconverter material Er³⁺, as introduced and discussed by Fischer et al. Using this approach, 1.48 W cm⁻² is equal to 493 suns, which we round to ~500 suns.

To clarify the approach for the reader, we now added this information in the caption of Fig. 3, line 156 – 158:

“Under one sun illumination, the irradiance in the absorption range of the upconverter Er³⁺ (1450 nm – 1600 nm) is 3 mW cm⁻² ⁵¹. “

Additionally, to make the definition of irradiance in suns easier for the reader, we changed the x-axis title in Fig. 4c, as well as Supplementary Figure S11 from “Incident irradiance [W/cm²]” to “Incident irradiance 1450 nm – 1600 nm [W cm⁻²]”.

51. Fischer, S., Fröhlich, B., Steinkemper, H., Krämer, K. W. & Goldschmidt, J. C. Absolute upconversion quantum yield of β -NaYF₄ doped with Er³⁺ and external quantum efficiency of upconverter solar cell devices under broad-band excitation considering spectral mismatch corrections. *Sol. Energy Mater. Sol. Cells* **122**, 197–207 (2014).

3. Previous studies have also observed non-linear scaling of higher absolute output powers with higher input powers, akin to a harmonic generation process. When upconversion is presented as a relative factor, as in this manuscript, it is not necessarily as obvious that this remains the case. Can the authors show how the absolute upconversion power should scale down from the intensity they measured down to an intensity of one sun, based on their model?

In the case of the upconversion photoluminescence, a higher irradiance (meaning the incident irradiance times the energy density enhancement in the Bragg structure) as input, does result in a higher upconversion photoluminescence (UCPL). In case of the upconversion quantum yield (UCQY), which is often shown in literature, the question whether the upconversion quantum yield increases or decreases with irradiance, depends on the considered irradiance level. To answer this question for the reader, we now included an additional Supplementary Figure S11 in the supplementary information showing the simulated UCPL and internal UCQY:

In the Supplementary information, we added a short discussion of Supplementary Figure S11:

“Fig. S11a shows the simulated UC photoluminescence (UCPL) and internal UC quantum yield (UCQY) of the main UC emission at 984 nm down to one sun irradiance. The simulated Bragg structure design is equivalent to the design for which the dependence of the relative UCPL on the irradiance is shown in Fig. 4c in the manuscript. Fig. S11a and Fig. S11b show the UCPL and UCQY, respectively, on a logarithmic scale, to visualize the lower irradiance regime, while in Fig. S11c, the UCQY is plotted on a linear scale, as mostly done in literature. The internal UCQY is defined as the number of emitted photons in the UC emission at 984 nm per absorbed photon⁹.

The UCPL increases with higher incident irradiances, such that the Bragg structure always outperforms the reference. The UCQY depicts a maximum. At irradiances above this maximum, other UC emissions become more likely that require the energy of more than two photons to participate in the UC process. This leads to a decrease in the UCQY of the 984 nm UC emission. Going towards lower irradiances, the photonic effects of the Bragg structure become exceedingly important compared to the reference. At one sun irradiance, the UCQY in the Bragg structure reaches 0.60%, while the reference only reaches 0.41%.”

9. Hofmann, C. L. M. *et al.* Enhanced upconversion in one-dimensional photonic crystals. A simulation-based assessment within realistic material and fabrication constraints. *Opt. Express* **26**, 7537 (2018).

Fig. S11 Simulated UC efficiency down to 1 sun irradiance, at an excitation wavelength of 1523 nm for reference and Bragg structure design as in Fig. 4c of the manuscript. a, UC photoluminescence (UCPL). b, internal UC quantum yield (UCQY) on a logarithmic scale. c, UCQY on a linear scale.

In the manuscript, we address the UCPL and UCQY shortly, by adding to line 269 – 273 in the manuscript: “In the low irradiance regime, in which the reference performs poorly, an increase in energy density, leads to a stronger absorption, which increases the probability of an energy transfer UC process to take place, resulting in a higher $UCPL_{rel}$. This becomes evident when looking at the absolute UCPL simulated down to one sun irradiance (Supplementary Fig. S11a). Consequently, also the UC quantum yield increases significantly at low irradiances (Supplementary Fig. S11b and Fig. S11c).”

4. If these values are much smaller than the efficiency of direct solar conversion, can the authors briefly comment on what additional improvements or modifications would be needed to make these relevant to improving champion cell efficiencies?

There are several aspects that can be optimized to significantly increase the impact of the photonic structure and overall upconversion. To answer this important question also for the reader, we now included a comprehensive answer in the discussion of the manuscript, line 342 – 362:

“There are several ways to further improve the efficiency of such photonic upconverter devices: i) increasing the number of layers in the Bragg structure largely increases the UC photoluminescence enhancement, e.g. from a factor of 4.4 at one sun irradiance for the layer stack we investigate in this manuscript with four active layers, to a factor of 66.6 for a layer stack with 20 active layers, considering the same production accuracy. ii) Using a material with a higher refractive index for the high-refractive index layer of the Bragg structure also increases the photonic effects⁵⁶. iv) Applying down-shifting materials for spectral concentration into the absorption range of the upconverter could increase the used spectral fraction as well as the irradiance acting on the upconverter and therefore increase the efficiency at lower irradiance levels^{57–59}. Superior UC properties have already been demonstrated for hybrid upconverter materials of lanthanide-doped upconverter nanoparticles combined with organic dyes as sensitizers, and have been applied to photovoltaic systems¹³. Down-shifting the complete spectral range below the bandgap of Silicon into the absorption range of the upconverter Erbium, is estimated to increase the current enhancement in the silicon solar cell by a factor of three in comparison to a purely Erbium-based system⁶⁰. v) Via rear-side mirrors, the irradiance in the photonic upconverter device could be further increased. vi) In addition, concentration optics would allow to operate in an irradiance regime in which the upconverter features a higher UC quantum yield, such as in conventional concentrator modules⁵² or in devices with concentrator optics specifically designed for UC⁶¹. With these measures combined, an optimistic estimate is to generate an additional current of 1.7 mA cm⁻² in a silicon solar cell⁶⁰. Especially in silicon-based tandem solar cells, in a situation where the silicon bottom-cell is limiting the overall current, this could have a significant impact on overall performance. To reach this goal, further progress and optimization in all mentioned areas is necessary. The contribution of this paper is to provide a validated theoretical model to enable a knowledge-based optimization process of photonic upconverter devices.”

13. Wen, S. *et al.* Future and challenges for hybrid upconversion nanosystems. *Nat. Photonics* **13**, 828–838 (2019).
52. Wiesenfarth, M., Anton, I. & Bett, A. W. Challenges in the design of concentrator photovoltaic (CPV) modules to achieve highest efficiencies. *Appl. Phys. Rev.* **5**, 41601 (2018).
56. Hofmann, C. L. M. *et al.* Enhanced upconversion in one-dimensional photonic crystals. A simulation-based assessment within realistic material and fabrication constraints. *Opt. Express* **26**, 7537–7554 (2018).
57. Strümpel, C., McCann, M., Del Canizo, C., Tobias, I. & Fath, P. Erbium-doped up-converters of silicon solar cells: assessment of the potential. *In Proceedings of the 20th European Photovoltaic Solar Energy Conference (EUPVSEC)*, 43-46 (2005).
58. Goldschmidt, J. C., *et al.* Advanced Upconverter Systems with Spectral and Geometric Concentration for high Upconversion Efficiencies. *In Proceedings of the conference on Optoelectronic and Microelectronic Materials and Devices (COMMAD)*, 1097-2137 (2008).
59. Goldschmidt, J. C., Löper, P. & Peters, I. M. Solarelement mit gesteigerter Effizienz und Verfahren zur Effizienzsteigerung DE102007045546 (2007).
60. Goldschmidt, J. C. *et al.* Record Efficient Upconverter Solar Cell Devices. *In Proceedings of the 29th European Photovoltaic Solar Energy Conference and Exhibition (EUPVSEC)*, 1-4 (2014).
61. Arnaoutakis, G. E. *et al.* Enhanced energy conversion of up-conversion solar cells by the integration of compound parabolic concentrating optics. *Solar Energy Materials & Solar Cells* **140**, 217–223 (2015).

5. If we compare to the modeling framework developed by the authors in prior work (e.g., C.L.M. Hofmann *et al.*, *Opt. Express* 26, 7537, 2018 and C.L.M. Hofmann *et al.*, *Opt. Express* 24, 14895, 2016), are there any significant changes? Section 3.10 seems to indicate that there are not many, but the upconversion graphs and trends (e.g., Figs. 2b, 3c, and 4) look significantly different.

The main focus of the paper Hofmann *et al.* 2018 (C.L.M. Hofmann *et al.*, *Opt. Express* 26, 7537, 2018)² was to investigate the photonic effects of a Bragg structure on upconversion, while varying the refractive index of the Bragg structures' high refractive index layer, the number of active layers as well as different production accuracies. The relative upconversion photoluminescence (UCPL) and upconversion quantum yield (UCQY) were analyzed. The simulation methods used are identical to the ones used in this manuscript. Therefore, the observed trends and conclusions are in agreement, however most graphs are not directly comparable because different parameters were investigated. In this paper, we now provide experimental evidence that the used simulation framework is valid.

The paper Hofmann *et al.* 2016 (C.L.M. Hofmann *et al.*, *Opt. Express* 24, 14895, 2016)³ allows for a more direct comparison, as it also contains the analysis of the impact of the two photonic effects, local density of optical states and energy density, separately on upconversion efficiency. The refractive indices of the investigated Bragg structures, however, are different: $n_{\text{low}} = 1.5$, $n_{\text{high}} = 1.8$ and the number of active layers is varied. The simulation methods used are slightly different, as pointed out in the publication of the more advanced version in Hofmann *et al.* 2018. Two changes are significant: i) in Hofmann *et al.* 2016 only ideal Bragg structures were investigated, no production accuracies are included. ii) Furthermore, there has been a small bug in the simulation script of the local density of optical states in Hofmann *et al.* 2016, which has an impact on the trends visible in the graphs discussed below. This bug was fixed in the version published in Hofmann *et al.* 2018 and we are currently working on an Erratum to the paper Hofmann *et al.* 2016 to correct the errors. The main findings that are discussed in Hofmann *et al.* 2016, however, are not influenced by this error and will not change in the Erratum.

In the following, we give a detailed figure by figure comparison to explain the exact differences:

Addressing Fig. 2b in this manuscript in comparison to Hofmann *et al.* 2018 and Hofmann *et al.* 2016:

Fig. 3b, published in Hofmann *et al.* 2018 can be compared to Fig. 2b in this manuscript. The refractive indices used for simulating the two graphs are similar: Fig. 3b in Hofmann *et al.* 2018 is simulated using round numbers for a simulation based analysis, $n_{\text{low}} = 1.5$, $n_{\text{high}} = 2.3$; Fig. 2b in this manuscript is simulated using experimentally determined refractive indices $n_{\text{low}} = 1.47$, $n_{\text{high}} = 2.28$. The main difference is the number of active layers simulated, in Fig. 3b in Hofmann *et al.* 2018, 25 active layers are considered, while in Fig. 2b in this manuscript, four active layers are considered. This has a large impact on the shape of the reflectance peaks as can be seen when comparing the two graphs.

Fig. 3. (a) Average relative energy density \bar{u}_{rel} across the active layers as a function of design wavelength λ_D for an exemplary Bragg structure with $n_{high} = 2.3$ and $\#_{al} = 25$. The upper x -axis indicates the active layer thickness, $d_{low} = \lambda_D / (4n_{low})$. (b) Reflectance, R , for the example structure at the design wavelength yielding the maximum \bar{u}_{rel} -value, λ_D^{max} (marked by a black, shaded circle in panel a). (c) Spatial energy density distribution inside the structure for $\lambda_D = \lambda_D^{max}$. Additionally, the refractive index profile is shown. (d) Average relative energy density across the active layers as a function of n_{high} and $\#_{al}$. The example structure considered in panels a, b and c is marked by a white, shaded circle.

Fig. 3 published in Hofmann et al. 2018 4.

Fig. 3c published in Hofmann *et al.* 2016 can also be compared to Fig. 2b in this manuscript. The differences are the refractive indices of $n_{low} = 1.5$, $n_{high} = 1.8$ Fig. 3c in Hofmann *et al.* 2016, shifting the position of the main reflectance peak. Additionally, the simulation is performed for nine active layers (in Hofmann *et al.* 2016 still called 20 bilayers including non-doped PMMA layers of half the thickness as the outer layers, reducing the reflectance at wavelengths lower than the main reflectance peak).

Fig. 3. a) Mean irradiance enhancement $\bar{\gamma}_I$ in the active layers of a 20 Bilayer Bragg stack, reaching a sharp maximum of $\bar{\gamma}_{I,max}$ at $\lambda_{D,max}$. b) Local irradiance $I(x)$ at each position x in the Bragg stack, plotted for the design wavelength $\lambda_{D,max}$. On the right y-axis the refractive index is shown. Active layers, containing the upconverter nanoparticles, are highlighted in grey. All maxima of $I(x)$ are positioned in the active layers, which leads to the high enhancement of $\bar{\gamma}_{I,max}$. c) Reflectance of the same optimized Bragg stack with a design wavelength $\lambda_{D,max}$. For this design, the excitation wavelength λ_{exc} lies right at the band edge of the photonic structure.

Fig. 3 published in Hofmann *et al.* 2016³.

Addressing Fig. 3c in this manuscript in comparison to Hofmann *et al.* 2016:

The before mentioned bug in evaluating the local density of optical states is visible in the graphs Fig. 6c and Fig. 6d published in Hofmann *et al.* 2016 in comparison to Fig. 3a and Fig. 3c of this manuscript. The different position and width of the bandgaps is due to the difference in refractive indices in Hofmann *et al.* 2016 and in this manuscript. Fig. 3a in this manuscript shows the corrected trend, while in Fig. 6c in Hofmann *et al.* 2016, the trend is not correct. However, the behavior inside the photonic bandgap is very similar in Fig. 6c in Hofmann *et al.* 2016 and Fig. 3a in this manuscript, which is the reason why the bug does not largely change the final effect on upconversion. This can be seen in the relative upconversion photoluminescence only taking into account the effect of the local density of optical states: this is displayed in Fig. 3c in this manuscript for two different irradiances and in the green curve in Fig. 6d in Hofmann *et al.* 2016. The green curve in Fig. 6d in Hofmann *et al.* 2016 is simulated for an irradiance of 0.59 W cm^{-2} (5890 W m^{-2}). This is in a similar irradiance regime as the 1.48 W cm^{-2} shown in the solid line in Fig. 3c in this manuscript. These two curves show a similar trend with differences caused by different considered refractive indices, a slightly different irradiance and the small effect of the bug in the simulation script of the local density of optical states.

Fig. 6. Change of UC performance, only regarding the effect of the LDOS by setting $\gamma_l(x)$ to unity. a) UCQY in dependence on design wavelength λ_D and incident irradiance I_{in} , reaching a maximum of 16.3%. b) UCQY in dependence on I_{in} (cut through graph (a) at $\lambda_D = 1632$ nm). The maximum UCQY of the Bragg stack (with $\gamma_l = 1$) is higher than that of the reference and is reached at a lower I_{in} . c) Mean relative LDOS for the main UC emission L31 (${}^4I_{11/2}$ to ${}^4I_{15/2}$) and loss mechanism L21 (emission ${}^4I_{13/2}$ to ${}^4I_{15/2}$). When the respective transition falls into the bandgap (highlighted region 1.BG), $\bar{\gamma}_{LDOS,if}$ is strongly reduced. d) Relative luminescence of L31 and L21 influenced by $\bar{\gamma}_{LDOS,if} \cdot \Gamma_{Lum,31}$ is enhanced in the region where $\bar{\gamma}_{LDOS,21}$ is suppressed. e) UCQY in dependence on λ_D (cut through graph (a) at $I_{in} = 5890$ W/m²). The UCQY within the Bragg stack follows the course of $\Gamma_{Lum,31}$. Due to the changed LDOS, the maximum possible UCQY within the Bragg stack is higher than that of the reference.

Fig. 6 published in Hofmann *et al.* 2016³.

Addressing Fig. 4 in this manuscript in comparison to Hofmann *et al.* 2016:

Fig. 4 in this manuscript contains the key novel results of this manuscript. Experimental results are presented as well as a variation of the excitation wavelength and irradiance, which was not addressed in the prior publications Hofmann *et al.* 2016 and Hofmann *et al.* 2018. Therefore, these figures cannot be compared to the earlier publications.

The only comparison that can be drawn is between the simulated curve in Fig. 4a in this manuscript and the green curve in Fig. 7e published in Hofmann *et al.* 2016. They both display the simulated relative upconversion photoluminescence of the main upconversion emission. The difference is the refractive indices used in the two examples (see above) as well as the number of active layers. Fig. 7e in published in Hofmann *et al.* 2016 shows the

example for 19 active layers (here called 40 bilayers), which, compared to the four active layers in Fig. 4a in this manuscript, increases and spectrally sharpens the enhancement peak. The impact of the bug in the calculation of the local density of optical states is hardly visible in Fig. 7e in Hofmann *et al.* 2016, because the effect of the energy density is much higher for a Bragg structure with as many as 19 active layers.

Fig. 7. Optimization of UC performance for an exemplary Bragg stack of 40 bilayers (BL). a) UCQY in dependence on design wavelength λ_D and incident irradiance I_{in} , reaching a maximum of 15.8%. b) UCQY in dependence on I_{in} (cut through graph (A) at $\lambda_D = 1604.5$ nm). Compared to the reference, the higher maximum UCQY of the Bragg stack is already reached at $I_{in} = 600$ W/m², mainly due to the high irradiance enhancement. D) Mean relative LDOS for the main UC emission L31 (${}^4I_{11/2}$ to ${}^4I_{15/2}$) and loss mechanism L21 (emission ${}^4I_{13/2}$ to ${}^4I_{15/2}$). E) Relative luminescence of L31 and L21 influenced by $\bar{\gamma}_I$ and $\bar{\gamma}_{LDOS,if}$. $\Gamma_{Lum,31}$ is mainly governed by $\bar{\gamma}_I$, rising non-linearly with an increasing $\bar{\gamma}_I$. F) UCQY in dependence on λ_D (cut through graph (A) at $I_{in} = 600$ W/m²). The UCQY within the Bragg stack is a superposition of the shapes of $\bar{\gamma}_I$ and $\bar{\gamma}_{LDOS,if}$, but again reaches a similar maximum as in Fig. 6. In conclusion, the increased maximum of the UCQY is due to the changed LDOS, while the shift of the maximum to shorter I_{in} is mainly caused by $\bar{\gamma}_I$.

Fig. 7 published in Hofmann *et al.* 2016³.

6. In Fig. 4a, the upconversion data has a fair spread. What is the main source of this spreading in the vertical direction at each design length?

To clarify our interpretation of the main source of this spreading, we adapted the old passage addressing this question in line 222 – 234:

“The variation of the measured values can be explained by slight thickness variations of the single layers in each stack. Most random thickness variations of single layers lead to a decrease in energy density in the active layers and therefore to a reduced UCPL_{rel}. For particular designs though, non-periodic thickness variations of single layers can

lead to an additional strong increase of the energy density in the active layers⁵³, which consequently leads to an additional increase in $UCPL_{rel}$. This might contribute to a maximum measured enhancement of 4.1 for $\lambda_{design} = 1844$ nm. In simulation, we also take the impact of the production inaccuracy on $UCPL_{rel}$ into account. However, the simulation features the mean expected reduction over 1000 separate calculations. A closer analysis of the impact of the non-periodicity within each single BS design, is out of scope of this paper.”

And included more relevant information, such that the passage now reads:

“We expect that the main reason for the variation of the single $UCPL_{rel}$ measurements, also within the same design wavelength, are slight thickness variations of the single layers in each stack that appear due to production inaccuracies (see Sections 3.1 and 3.2). Despite these thickness variations of single layers, a defined design wavelength can be assigned to each sample we investigated (Supplementary Figure S7). In simulation, we also take the impact of the production inaccuracy on $UCPL_{rel}$ into account. However, the simulation features the mean expected reduction over 1000 separate calculations. Most random thickness variations of single layers lead to a decrease in energy density in the active layers and therefore to a reduced $UCPL_{rel}$. For particular designs though, non-periodic thickness variations of single layers can lead to an additional strong increase of the energy density in the active layers⁵³, which consequently leads to an additional increase in $UCPL_{rel}$. This might contribute to a maximum measured enhancement of 4.1 for $\lambda_{design} = 1844$ nm. A closer analysis of the impact of the non-periodicity within each single Bragg structure design, is out of scope of this paper.”

53. Spallek, F., Buchleitner, A. & Wellens, T. Optimal trapping of monochromatic light in designed photonic multilayer structures. *J. Phys. B: At. Mol. Opt. Phys.* **50**, 214005 (2017).

7. Also, given the current data in Fig. 4a, how confident can one be about the simulated peak conversion taking place with a design wavelength around 1850 nm (and an active layer thickness around 630 nm)?

The experimental data in Fig. 4a in the manuscript follows the trend from simulation, however, as you pointed out, it is not sufficient to validate the simulated peak at 1855 nm design wavelength. However, the excitation wavelength dependent analysis shown in Fig.4b in the manuscript gives further evidence that wavelength dependent effects are described correctly.

We adapted our explanation of the experimental results of Fig. 4a to clarify which findings are theoretical and which are experimental at this point, by adapting line 217 – 220 from the old formulation:

“As expected from simulation, in experiment the photonic effects increase the UC signal for λ_{design} around the expected maximum $UCPL_{rel}$ at 1855 nm (group II at the maximum and group III close to the maximum).”

It now reads:

“In experiment the photonic effects increase the UC signal for λ_{design} around the simulated maximum $UCPL_{rel}$ at 1855 nm. In group II and III, at and close to the simulated maximum, respectively, the highest mean measured $UCPL_{rel}$ is found, while group III slightly outperforms group II.”

In line 245 – 248 we state that only with the data shown in Fig. 4b, we are confident to talk about a very good agreement between simulation and experiment: “The slope expected from simulation, which characterizes the Bragg structures effects, is very well visible in the experimental data in all five groups. We performed the same evaluation for the UC emission around 814 nm (Supplementary Fig. S12) and found the same good agreement between simulation and experiment.”

So we can say, with the good agreement of slopes in Fig. 4b in the manuscript, we are confident that the peak conversion takes place around the design wavelength of 1855 nm, as shown in the simulation in Fig. 4a in the manuscript.

Your comment about the optimized active layer thickness being about 630 nm revealed a little mistake we made in the top x-axis of Fig. 4a. We now corrected this mistake and added a short description of the top x-axis in line 214 - 215 :

“Both, the active- and TiO₂ layer are scaled to match the desired design wavelength λ_{design} . The corresponding active layer thickness is shown in the top x-axis of Fig. 4a.”

8. 8a) In the discussion, since some of the prior results take place over a narrow angular range, is it possible to estimate the photovoltaically-relevant enhancement for each system (including this one) at equal intensities? Then you would have the option of presenting a table summarizing the upconversion efficiency associated with each experimental photonic upconversion structure.

We understand that your question has two parts: The first part is referring to the angular characteristics, the second part to an overview of literature results.

In the answer to question 2 of reviewer 2, we are extensively discussing angular behavior and the implication for photovoltaics and we have added a graph showing the angular characteristics for incident radiation, which hopefully is a sufficient answer for the first part of your question. Please refer to question 2 of reviewer 2 for details.

For the second part of your question, we agree that such an overview would be very helpful. There are, however, several difficulties, why we cannot provide such an overview within the scope of this paper. First, you are right that for a fair comparison, the intensity would need to be taken into account. However, as you can see from Figure S11, the dependence of the upconversion photoluminescence and the upconversion quantum yield are not simply quadratic, or linear, respectively. That is, also scaling, e.g. dividing the upconversion quantum yield by the irradiance, does not yield a fair comparison. Typically, the highest upconversion quantum yields are measured for very high irradiance, but even for the same material, the upconversion quantum yield divided by the irradiance drops with the irradiance⁵.

Furthermore, the dependence on the irradiance critically depends on the upconverter material, which also complicates the assessment of the effect of photonic structures. A bad upconverter material, which is a material with a lot of additional recombination, could strongly profit from a photonic irradiance enhancement, as higher energy levels are more populated and energy transfer upconversion becomes more likely. On the other hand, a better material with less recombination, and thus already more occupied higher energy levels, could see a decrease in upconversion quantum yield because more three-photon and more-photon upconversion processes are triggered by the same photonic structure. A simple rule of thumb is that you see higher enhancement factors when you choose a less efficient upconverter.

Finally, the angular characteristics of different photonic-upconverter structures are different and so are measurement conditions:

- i) detection of only a small angular fraction or detection with an integrating sphere;
- ii) illumination from a specific angle or screening of a full range;
- iii) choice of the reference sample that the upconversion enhancement factor is compared to;
- iv) the irradiance that a measurement is performed at;

Correcting for these differences would require to simulate the angular characteristics of all considered structures. In our opinion, for such an overview table the community would need to agree to a certain set of standard measurement conditions, which are applied to all samples.

8b) That would make it much more clear whether this particular 1D photonic crystal has any special properties in its own right, or if the advance is primarily in validating the model.

Bragg structures have many favorable qualities for the purpose of our investigation: the detailed comparison of simulation and experiment. Additionally, they have many features that are important as well as promising for an application in photovoltaics. Thank you for bringing up that this point needs more explanation. To clarify for the reader why we chose to look at Bragg structures and what our findings are for this particular structure, we added all important points in two passages, one in the introduction, one in the discussion of our manuscript:

In the introduction, we changed line 91 - 95:

“We choose to investigate a simple Bragg structure (BS) design that allows for revealing the essential aspects and yet is also application relevant as it could be fabricated on an industrial scale. The structure consists of alternating quarter wave layers “

to: “We choose to investigate a simple Bragg structure design that reveals the essential aspects and is also application relevant (it could be fabricated on an industrial scale). Another key advantage of a layer stack system is that it is possible to add many layers, so the overall volume of upconverter material on which the photonic structure acts can be large. The overall absorption can therefore be high, unlike in other systems, where high enhancements are confined to very small volumes.”

And in the discussion, we changed the paragraph in line 297 – 316:

“The BS we investigated in this work is flexible in various properties and therefore promising for UC enhancement. The amount of upconverter material is not limited by the design but can be adapted by adding more layers to the stack. UCPL enhancement occurs in a broad spectral range, covering most of the investigated Er^{3+} absorption range. This is important for broad-band applications, such as photovoltaics. We chose to investigate a simple BS with only 4 active layers to be able to tune and understand all appearing effects. The production accuracy we reached in experiment allowed for a detection of the full expected photonic effects.”

To: “We chose to investigate a simple Bragg structure with only 4 active layers to be able to tune and understand all appearing effects. The production accuracy we reached in the experiments allowed for a detection of all the expected photonic effects. Even though Bragg structures might not be the photonic structures showing the highest UC enhancement factors, they have many features that are important and promising for UC enhancement for an application in photovoltaics: The amount of upconverter material, and thereby the overall absorption, is not limited by the design but can be adapted by adding more layers to the stack. Furthermore, as we could show, UCPL enhancement occurs in a broad spectral range, covering most of the investigated Er^{3+} absorption range. We additionally investigated the relative UCPL under a varied incident angle both theoretically and experimentally. The analysis is shown in the Supplementary Section 4.5, where the very good agreement of theory and experiment is documented. We find that light can be efficiently coupled into the structure up to large incident angles of about 30° (Supplementary Figure S13). This is important for broad-band, wide-angle applications like photovoltaics: for a simple system without tracking, the movement of the sun means varying incident angles, but also for concentrator systems using tracking, the concentration means that the angular range of the light incident onto the solar cell is increased. In conclusion, our analysis showed that a Bragg structure has spectral and angular characteristics that are beneficial for the application in photovoltaics.”

Reviewer #2 (Remarks to the Author):

In this manuscript, the authors investigate the effects of 1D-photonic structure on photon UC in embedded upconverter nanoparticles through both simulation and experimentation. Their results show the experimental validation of a comprehensive simulation modeling framework. By studying the parameters of active layer thickness, irradiance, and excitation wavelength, the authors prove that experimentally measured UC enhancement features the expected values and behavior from the simulation in all three performed parameter scans, reaching $82\pm 24\%$ of the simulated UC enhancement out of 2480 measurements with different parameter combinations. In fact, in the past decades, the effect of photonic crystals on the luminescence of embedded light-emitters has been extensively studied, both experimentally and theoretically. For example, GdVO₄:Eu³⁺ nanophosphors were sandwiched between two photonic multilayers based on transparent ZrO₂ and SiO₂ layers to increase the needed emission (Nanoscale Horizon, DOI: 10.1039/c8mh00123e). Many works have utilized photonic crystals to extract the emission of LED. I feel this work lacks the novelty for Nature Comm. A few other comments for improvement:

The paper by Geng *et al.* (DOI: 10.1039/c8mh00123e)⁶ is a very interesting and careful study of a tuned optical cavity structure to influence both chromaticity and directionality of the emitted light of integrated nanophosphors. The manuscript by Geng *et al.* aims at an application in nanoscale photonics based solid state lighting and involves simulations of reflectance and electric field intensity. Thank you for pointing out this work, we included it in our literature overview in the introduction, line 75 – 76:

“Also detailed understanding of plasmonic enhancement effects are of major interest in various areas of application^{46,47}.”

46. Kravets, V. G., Kabashin, A. V., Barnes, W. L. & Grigorenko, A. N. Plasmonic Surface Lattice Resonances: A Review of Properties and Applications. *Chem Rev* **118**, 5912–5951 (2018).

47. Geng, D., Cabello-Olmo, E., Lozano, G. & Míguez, H. Photonic structuring improves the colour purity of rare-earth nanophosphors. *Mater. Horiz.* **5**, 661–667 (2018).

However, in comparison to Geng *et al.*, our manuscript adds significant novel aspects: We do not only look at electric field intensities, but in our simulation model additionally include the local density of optical states. With these two effects, we take into account all important effects of the photonic structure. Furthermore, we integrate the simulation of photonic effects into a comprehensive modeling framework of the upconversion dynamics.

In comparison to Geng *et al.*, who focus on down-shifting materials for solid state lighting, we investigate upconverter materials for photovoltaics. Our focus involves additional challenges:

i) currently, efficient upconverter materials exhibit lower quantum yields than efficient nanophosphors and performing measurements with low signals is more challenging.

ii) We aim at an application in photovoltaics. Measuring systems at very low irradiances in a regime which is relevant for photovoltaics is a challenge, while for applications in solid state lighting, investigations can be done at arbitrary irradiances.

iii) Most importantly, the non-linear dependence of upconversion on the irradiance needs to be addressed very carefully. We do this by comprehensively describing the upconversion dynamics in our simulation model.

To our knowledge, we present the first study with a comprehensive simulation model including photonic effects and upconversion dynamics and an experimental validation of the model including large parameter scans as well as large statistics with 2480 single measurements. With this work, we contribute to the field a theoretical understanding and an experimentally validated approach to theoretically optimize photonic structures for upconversion enhancement.

We emphasize that the full potential of photonic structures can only be exploited when a theoretical optimization of the photonic-upconverter system is performed.

1. For 1D photonic structure, reflection spectrum is dependent on the incident angle. In this work, what angle was the reflection spectrum measured?

A detailed discussion of the spectrometer measurements and determination of the design wavelength is given in the supplementary information, Section 3.1. In this section, the angle, at which reflection measurements are performed, is given: "We performed the measurement in an integrating sphere with a tilt of the sample of 8°."

Thank you for pointing out that this relevant information was not included in the shortened version of the measurement details in the manuscript itself so far. We now also included it in the methods section of the manuscript, adding to line 409 - 410:

"With a spectrophotometer (Lambda 950, PerkinElmer, Germany) we measured the characteristic **BS Bragg structure** reflectance for each sample point **at a tilt of 8° relative to the incident beam.**"

Additionally, in response to your second question, we included a new Supplementary Figure S13. Thereby, Supplementary Figure S13a shows how the reflectance of three exemplary Bragg structure designs changes with a varied incident angle. Please refer to our answer to question 2 for details.

2. The upconversion photoluminescence is measured in an integrating sphere. Due to the inherent directional nature of photonic crystals, the enhancement effect should also be directional. The authors should compare the integrated effect and one-direction effect by simulation and experiments.

Yes, the enhancement effects on an embedded upconverter are expected to be strongly directional in photonic structures. As this is a very important question, we now included an experimental and theoretical analysis of the effect of a varied incident angle on upconversion photoluminescence. The results are plotted in the Supplementary Figure S13 and a detailed description of the experimental setup, measurement details and interpretation of results is given in a new Supplementary Section 4.5. Please refer to Supplementary Section 4.5 for details.

In the discussion of the manuscript we included the essence of our investigation and discuss our findings with respect to an application in photovoltaics. Line 305 – 313:

"We additionally investigated the relative UCPL under a varied incident angle both theoretically and experimentally. The analysis is shown in the Supplementary Section 4.5, where the very good agreement of theory and experiment is documented. We find that light can be efficiently coupled into the structure up to large incident angles of about 30° (Supplementary Figure S13). This is important for broad-band, wide-angle applications like photovoltaics: for a simple system without tracking, the movement of the sun means varying incident angles, but also for concentrator systems using tracking, the concentration means that the angular range of the light incident onto the solar cell is increased."

Directionality of emission is the other part of the angle dependence. We already designed our simulation tool to be able to integrate the theory for directionality of emission as a fractional local density of optical states. This, however, will be subject of our future work. At this point, we just would like to point out that for the targeted application in photovoltaics, all the light leaving the upconverter into the half-face towards the solar cell will enter the solar cell, thus angular dependence of the emission is of less concern.

We describe this in line 294 – 296: "... this method allows for an investigation of directionality of UC emission, implemented as a fractional LDOS⁵⁴, which will be subject of our future work."

54. Gutmann, J., Zappe, H. & Goldschmidt, J. C. Quantitative modeling of fluorescent emission in photonic crystals. *Phys. Rev. B* **88** (2013).

Fig. S13, Investigation of the UC photoluminescence ($UCPL_{rel}$) in different Bragg structure designs for a spectrally and angle resolved excitation. a, simulated reflectance for three different experimentally investigated Bragg structure designs

to illustrate the shift of the photonic bandgap. The maximum design wavelengths measured on each sample is displayed (i-iii). b, simulated reflectance, zoomed in into the experimentally investigated spectral range. c, simulated UCPL_{rel} for the maximum design wavelengths measured. The maximum UCPL_{rel} shifts along with the photonic band edge towards smaller wavelengths at higher angles. d, simulated UCPL_{rel} for the minimum design wavelengths measured. E, measured UCPL_{rel} on three different experimentally investigated Bragg structure designs. The measured range of design wavelengths on each sample is indicated in the graph. The trend for each sample in comparison to the simulation of the maximum (c) and minimum (d) design wavelengths measured, is in good agreement. Efficient incoupling of light is possible out of a large angle range, up to about 30° (for design ii).

3. It is well known the PMMA has an absorption in the NIR region. This effect needs to be taken into consideration.

In Fig. 1 you can find the absorptance in a $1 \mu\text{m}$ thin PMMA layer, which approximately is the total thickness of the active layers in the Bragg structures we investigated in this work. It is calculated via the absorption coefficient extracted from the spectrophotometer measurement of a 4.9 mm thick PMMA sample. Within the absorption range of the upconverter around 1523 nm , PMMA depicts a negligible absorption of maximally $1.4 \cdot 10^{-4}$ in the complete absorption range and maximally $2.1 \cdot 10^{-5}$ in the range that we performed our measurements in. In the emission range of the upconverter around 984 nm and 814 nm , the PMMA absorptance is also negligible. Therefore, for our experiments, the PMMA absorptance does not have a significant influence. However, if the number of layers within the stack is significantly increased, one might want to consider using a different polymer.

Fig. 1, Absorptance in a $1 \mu\text{m}$ thin PMMA layer; normalized UC photoluminescence and absorptance in the upconverter NaYF_4 doped with 25% Er^{3+} for the transitions investigated in this manuscript.

4. The thicknesses of high and low refractive index materials determine the bandgap of photonic crystals. This work only investigated the effect of active layer thickness. The thickness of TiO_2 should also be considered.

In this work, we investigate the effect of the design wavelength, which defines the optical thickness of both layers in the Bragg structure, line 124 – 126 in the manuscript: “The position of the reflectance peak, and therewith the first photonic bandgap, is determined by the design wavelength (λ_{design}) that defines the thickness $d_i = \lambda_{\text{design}}/4n_i$ of each layer i with refractive index n_i .”

However, we only mentioned this in the very beginning of the results section. To clarify for the reader, we added an explanation in the discussion of Fig. 4a, line 214 – 215:

“Both, the active- and TiO₂ layer are scaled to match the desired design wavelength λ_{design} . The corresponding active layer thickness is shown in the top x-axis of Fig. 4a.”

The impact of the high refractive index layer in the Bragg structure indeed plays a major role. We have investigated this important aspect by varying the refractive index and analyzing the resulting photonic effects on upconversion in a simulation based approach⁴.

Reviewer #3 (Remarks to the Author):

The manuscript by Hofmann and co-workers reports a very careful study on the effect of Bragg structures on rare-earth upconversion. They show that their experimental realisations come close to theoretical expectations, validating both.

Upconversion is an area of intense interest, and the recent review by Jin and others in Nature Photonics exemplifies this. The photonic approach to upconversion enhancement is one that demands careful reports of the kind presented here.

The paper is well written, and the reference list appears appropriate, with the exception perhaps of the above review. If I were to find one criticism, it is that I do not find the absolute upconversion efficiencies which satisfy the reader that progress is being made in the field. Several of the authors are thought leaders in this regard, so it is surprising that it is missing.

Otherwise, I would recommend publication as is, subject to minor language tweaks by the editorial process.

Thank you very much for acknowledging the way we present our work in this manuscript.

And thank you for pointing out the recent review by Wen, Jin and co-authors. We included this important publication in the literature overview in our introduction, line 53 – 56:

“Extensive research has been done on understanding the theory of the UC process¹⁻⁴ and on material development, predominantly nanocrystals⁵⁻¹⁵. By now, UC is exploited in a broad range of applications ranging from bioimaging^{13,16-20}, theranostics²¹⁻²⁴, security^{20, 25, 26}, data storage²⁷, and data analysis²⁸ to photovoltaics^{13,15,19,20,29,30}.”

As well as in the discussion, in line 349 - 351:

“Superior upconversion properties have already been demonstrated for hybrid upconverter materials of lanthanide-doped UCNPs combined with organic dyes as sensitizers, and have been applied to photovoltaic systems¹³.”

13. Wen, S. *et al.* Future and challenges for hybrid upconversion nanosystems. *Nat. Photonics* **13**, 828–838 (2019).

(not all references listed in this answer)

You are asking about the upconversion quantum yield as a measure of progress: In their review article, Wen, Jin and co-authors also discuss the need for a standardization of reported results in the different communities in order to enable a comparison. They suggest the upconversion quantum yield as a standardized measure.

The particular goal of our work, presented in this manuscript is to investigate upconversion enhancements due to photonic structure effects for an application in photovoltaics. To be able to make a statement about photonic effects in a low irradiance regime close to one sun, we measure at low intensities. Additionally, the simple structures we investigate for validating our model contain very thin layers and therefore little upconverter material. Measuring a reliable upconversion quantum yield at these low intensities and the small amount of upconverter material was, unfortunately, not possible with our setup. However, as discussed in the answer to question 3 of the first reviewer, we can say that both, the upconversion quantum yield and upconversion photoluminescence can be enhanced with a tuned Bragg structure at low intensities of one sun. Please refer to our answer to question 3 of reviewer 1 for details.

To discuss approaches how to increase the overall upconversion performance for the photonic-upconverter we investigated, we added a comprehensive passage in the end of our discussion in line 342 – 362. Please refer to our answer to question 4 of reviewer 1 on the same topic.

References

1. Wiesenfarth, M., Anton, I. & Bett, A. W. Challenges in the design of concentrator photovoltaic (CPV) modules to achieve highest efficiencies. *Appl. Phys. Rev.* **5**, 41601; 10.1063/1.5046752 (2018).
2. Hofmann, C. L. M. *et al.* Enhanced upconversion in one-dimensional photonic crystals. A simulation-based assessment within realistic material and fabrication constraints. *Opt. Express* **26**, 7537; 10.1364/OE.26.007537 (2018).
3. Hofmann, C. L. M., Herter, B., Fischer, S., Gutmann, J. & Goldschmidt, J. C. Upconversion in a Bragg structure: photonic effects of a modified local density of states and irradiance on luminescence and upconversion quantum yield. *Opt. Express* **24**, 14895–14914; 10.1364/OE.24.014895 (2016).
4. Hofmann, C. L. M. *et al.* Enhanced upconversion in one-dimensional photonic crystals. A simulation-based assessment within realistic material and fabrication constraints. *Opt. Express* **26**, 7537–7554; 10.1364/OE.26.007537 (2018).
5. Goldschmidt, J. C. & Fischer, S. Upconversion for Photovoltaics - a Review of Materials, Devices and Concepts for Performance Enhancement. *Advanced Optical Materials* **3**, 510–535; 10.1002/adom.201500024 (2015).
6. Geng, D., Cabello-Olmo, E., Lozano, G. & Míguez, H. Photonic structuring improves the colour purity of rare-earth nanophosphors. *Mater. Horiz.* **5**, 661–667; 10.1039/c8mh00123e (2018).

REVIEWER COMMENTS

Reviewer #1 (Remarks to the Author):

The response letter addressed the vast majority of my prior concerns. My follow-up responses and requests are as follows:

1. The responses to Reviewer 1 Questions 3 and 8a are in tension. While the authors indicate in Question 3 that upconversion can be modeled as a function of intensity, it appears less possible in response to Question 8 when a comparison table is requested. The difference in measurements is perhaps a stronger reason that the latter cannot be provided. However, some sort of comparison is needed to demonstrate whether a real improvement is available. If a full table is beyond the scope, the most comparable prior experiment would be an appropriate point of comparison.
2. The response to Reviewer 1 Question 8b should be elaborated a bit more to quantify the benefit of the 1D Bragg stack approach in terms of the increase in total upconverted power, given a reasonable set of assumptions, compared to other prior structures investigated, which may have 2D or 3D structures with much smaller modal volumes.
3. The response to Reviewer 1 Question 5 should be extended to include a brief discussion in the manuscript of the model update from CLM Hoffman 2016 to CLM Hoffman 2018, with a note that the latter model is employed under somewhat different conditions in this paper than it was previously (to explain the remaining apparent differences in the performance).

Thank you very much.

Reviewer #2 (Remarks to the Author):

I think all concerns raised by referees have been clearly addressed in this revised version. Although the overall enhancement performance is small (<10 folds), it is a good start for the community to realize the possibility of enhancing upconversion-mediated solar cells with simulation guidance. I thus recommend publication as is.

One comment for authors' consideration is that despite the promise of upconversion, a very small portion of NIR light can penetrate through currently available commercial solar cell panels.

Reviewer #3 (Remarks to the Author):

I have read the very detailed responses to the reviewers' comments. As one who was in favour of the manuscript, essentially "as is", in the initial stage, my approval of the manuscript is only strengthened. The manuscript should be published as amended.

We would like to thank all reviewers for their positive feedback on our revised manuscript, as well as for the suggestions for improvement from Reviewer 1. Based on these suggestions, we now included a comprehensive table in the Supplementary information in which we summarize design and experimental parameters for a selection of literature reports on photonic upconverter devices. Please find our detailed point-by-point response to all comments and questions raised below.

The *reviewer's comments and questions* are marked as italic text. Passages we **deleted** in the manuscript are highlighted in red, **new** passages we took in are highlighted in blue. Please also find a new uploaded version of the manuscript and supporting information with highlighted changes, as well as the source data file and the simulation code.

Reviewer #1 (Remarks to the Author):

The response letter addressed the vast majority of my prior concerns. My follow-up responses and requests are as follows:

1. The responses to Reviewer 1 Questions 3 and 8a are in tension. While the authors indicate in Question 3 that upconversion can be modeled as a function of intensity, it appears less possible in response to Question 8 when a comparison table is requested. The difference in measurements is perhaps a stronger reason that the latter cannot be provided. However, some sort of comparison is needed to demonstrate whether a real improvement is available. If a full table is beyond the scope, the most comparable prior experiment would be an appropriate point of comparison.

It is correct that with our simulation tool, we can model upconversion as a function of intensity. We included such a simulation in our response to Question 3 of Reviewer 1 in the first resubmission for the material system and photonic structure that we are investigating in this manuscript.

However, the used rate-equation model describing the upconversion processes is specific for the upconverter material, in our case β -NaYF₄:Er³⁺. Even when the same Er³⁺ ions are used as upconverter species, a different crystalline host changes the precise positions of energy levels, their spectral width, and also the magnitude of non-radiative processes¹. In consequence, considerable material characterization is necessary to obtain the relevant modelling parameters for each material. So, while it is possible for us to model the intensity dependence for our upconverter material, it is not possible to do so for the different materials used in the literature. These differences come on top of the variety in measurement conditions, which makes a comparison difficult (as argued in our response to Question 3 of Reviewer 1 in the first resubmission). To offer the best comparison possible, we now included a comprehensive overview table (Supplementary Table S1 – S4) in the Supplementary information for the selection of publications we cited in the introduction on photonic structure enhanced upconversion. The table summarises important parameters on upconverter material, photonic structure, various measurement details and performed simulations. For the reader, this gives an overview of the variety of investigations, and also why it is difficult to compare them. We attached the tables in this document on page 4-7.

We now refer to this summary table in the introduction when discussing enhancement factors reached in literature, line 75 – 76: **"A detailed overview can be found in the Supplementary Tables S1 – S4."** As well as in the discussion in line 322 – 323: **"In the Supplementary Tables S1 to S4, we provide an overview of design and experimental details for the photonic upconverter devices from literature discussed in the introduction."**

As you can see from the table, simulations beyond reflectance or transmittance properties are rarely performed. Some reports take into account the local density of optical states. No other works take into account the upconversion dynamics. Thus, our work presents a real progress to the field, as it provides an exhaustive theoretical framework, which is validated by experiments. This achievement is independent from the question, whether Bragg stacks are a superior type of photonic structure compared to others. However, we address the question of real improvement in our answer to question 2 below.

2. The response to Reviewer 1 Question 8b should be elaborated a bit more to quantify the benefit of the 1D Bragg stack approach in terms of the increase in total upconverted power, given a reasonable set of assumptions, compared to other prior structures investigated, which may have 2D or 3D structures with much smaller modal volumes.

In the Supplementary Table S1-S4, we also included the amount of upconverter material in each structure and whether or not this amount is limited by the design. High enhancement factors occur at the surface of 2D or 3D photonic structures and are thus limited to small volumes. The thickness of the affected material ranges from 200 nm to maximally 1.3 μm . These structures reach enhancement factors of ~ 30 or maximally 130.

Enhancement factors in layer stack designs are usually lower, reaching up to ~ 25 (strongly dependent on the number of layers and production accuracy²). The benefit of a layer stack design, as in our Bragg structure and also the works by Johnson *et al.* and Rojas-Hernandez *et al.*, is that the amount of upconverter material is not limited by the design. Our current designs feature a total thickness of the upconverter layers of $\sim 1.3 \mu\text{m}$. When adding more layers to the stack, the amount of upconverter material increases and also the upconversion enhancement factor increases as the photonic effects are stronger². In the report of Rojas-Hernandez *et al.*, a similar total amount of upconverter material is reported with a summed up thickness of $\sim 1 \mu\text{m}$, while Johnson *et al.* report on a layer stack with 60 layers, reaching $\sim 15 \mu\text{m}$ thickness in total.

As the amount of upconverter material is indeed an important parameter of a photonic structure, we added to our comparison to these two works in line 327 – 342:

“Johnson *et al.* also investigated a Bragg structure of Er³⁺-doped porous silicon. Under 1550 nm excitation and a high irradiance, they report a 26.5 and 5-fold enhancement of the green and 980 nm UC emission, respectively, for a structure similar to what we define a 30 active layer structure⁴⁶ (see Supplementary Table S4). The enhancement occurs in an incident angle range of approximately 4°. They mention difficulties in controlling the layer thickness, which crucially diminishes the photonic effects. This report agrees well with our simulation, including the correct refractive indices and a large layer thickness variation (discussed in⁵⁰) and pronounces the importance of including fabrication inaccuracy: a precise 4-active layer stack can reach an effect close to an imprecise 30-active layer stack. **The amount of upconverter material, of course, also needs to be considered: while the design used by Johnson *et al.* features a total thickness of all upconverter-doped layers of as much as $\sim 15 \mu\text{m}$, our investigated Bragg structures with four active layers features a summed up active layer thickness of $\sim 1.3 \mu\text{m}$.** Rojas-Hernandez *et al.* report a 25-fold enhancement of green UCPL under 975 nm excitation in a microcavity structure of 21 layers of TiO₂ and Tb³⁺/Yb³⁺-doped aluminosilicate glass, **featuring a summed up active layer thickness of $\sim 1 \mu\text{m}$** , measured at a distinct detection angle⁴⁴ (see Supplementary Table S3). In comparison, for a Bragg structure with 10 active layers (in total 21 layers) in our current production accuracy, from simulation we expect a UCPL enhancement of a factor of 4.3 at a relatively high irradiance of 1.48 W cm⁻² and 27 at a low irradiance of one sun.”

44. Rojas-Hernandez, R. E., Santos, L. F. & Almeida, R. M. Photonic crystal assisted up-converter based on Tb³⁺ / Yb³⁺ - Doped aluminosilicate glass. *Opt. Mater.* **83**, 61–67 (2018).

46. Johnson, C. M., Reece, P. J. & Conibeer, G. J. Theoretical and experimental evaluation of silicon photonic structures for enhanced erbium up-conversion luminescence. *Sol. Energy Mater. Sol. Cells* **112**, 168–181 (2013).

50. Hofmann, C. L. M. *et al.* Enhanced upconversion in one-dimensional photonic crystals. A simulation-based assessment within realistic material and fabrication constraints. *Opt. Express* **26**, 7537 (2018).

Currently, 2D and 3D photonic structures outperform 1D structures in terms of the total upconverted power (when assuming that the efficiency of the upconverter material itself would be the same in the structures that are compared). Therefore, our main motivation to investigate a Bragg structure was not to reach the highest possible upconversion enhancement factor, but to investigate a simple structure that allowed for validating the simulation tool because of its simplicity.

3. *The response to Reviewer 1 Question 5 should be extended to include a brief discussion in the manuscript of the model update from CLM Hofmann 2016 to CLM Hofmann 2018, with a note that the latter model is employed under somewhat different conditions in this paper than it was previously (to explain the remaining apparent differences in the performance).*

Thank you very much.

We included the discussed differences in the methods Section 3.10 in line 493 – 500:

“Compared to the current model version, the simulation methods used in Hofmann et al. 2016⁶⁸ were slightly different, as pointed out in the publication of the more advanced version in Hofmann et al. 2018⁵⁰. Two changes are significant: i) in Hofmann et al. 2016 only ideal Bragg structures were investigated, no production accuracies are included. ii) Furthermore, there has been a small bug in the simulation script of the local density of optical states in Hofmann et al. 2016, which has an impact on the trends visible in the graphs including the effect of the LDOS. This bug was fixed in the version published in Hofmann et al. 2018 and we are currently working on an Erratum to the paper Hofmann et al. 2016 to correct the errors. The main findings that are discussed in Hofmann et al. 2016, however, are not influenced by this error and will not change in the Erratum.”

50. Hofmann, C. L. M. et al. Enhanced upconversion in one-dimensional photonic crystals. A simulation-based assessment within realistic material and fabrication constraints. *Opt. Express* **26**, 7537 (2018).
68. Hofmann, C. L. M., Herter, B., Fischer, S., Gutmann, J. & Goldschmidt, J. C. Upconversion in a Bragg structure: photonic effects of a modified local density of states and irradiance on luminescence and upconversion quantum yield. *Opt. Express* **24**, 14895–14914 (2016).

Reviewer #2 (Remarks to the Author):

I think all concerns raised by referees have been clearly addressed in this revised version. Although the overall enhancement performance is small (<10 folds), it is a good start for the community to realize the possibility of enhancing upconversion-mediated solar cells with simulation guidance. I thus recommend publication as is.

One comment for authors' consideration is that despite the promise of upconversion, a very small portion of NIR light can penetrate through currently available commercial solar cell panels.

Thank you very much for this positive feedback. Indeed the transmission of NIR light is an issue. However, in³ and⁴ we have shown that in principle it is possible to achieve good transmission.

Reviewer #3 (Remarks to the Author):

I have read the very detailed responses to the reviewers' comments. As one who was in favour of the manuscript, essentially "as is", in the initial stage, my approval of the manuscript is only strengthened. The manuscript should be published as amended.

Thank you very much for this positive feedback.

References

1. Fischer, S. et al. Upconversion Quantum Yield of Er³⁺-doped β -NaYF₄ and Gd₂O₃. The Effects of Host Lattice, Er³⁺ Doping, and Excitation Spectrum Bandwidth. *Journal of Luminescence* **153**, 281–287; 10.1016/j.jlumin.2014.03.047 (2014).
2. Hofmann, C. L. M. et al. Enhanced upconversion in one-dimensional photonic crystals. A simulation-based assessment within realistic material and fabrication constraints. *Opt. Express* **26**, 7537; 10.1364/OE.26.007537 (2018).
3. Fischer, S., Favilla, E., Tonelli, M. & Goldschmidt, J. C. Record efficient upconverter solar cell devices with optimized bifacial silicon solar cells and monocrystalline BaY₂F₈. 30% Er³⁺ upconverter. *Solar Energy Materials & Solar Cells* **136**, 127–134; 10.1016/j.solmat.2014.12.023 (2015).
4. Rüdiger, M. et al. Bifacial n-type silicon solar cells for upconversion applications. *Solar Energy Materials & Solar Cells* **128**, 57–68; 10.1016/j.solmat.2014.05.014 (2014).

		opal PCs	opal PCs	opal PCs
Publication		Shi, Y. et al. Upconversion fluorescence enhancement of NaYF ₄ :Yb/Re nanoparticles by coupling with SiO ₂ opal photonic crystals. J. Mater. Sci. 54 , 8461–8471 (2019)	Niu, W. B. et al. 3-Dimensional photonic crystal surface enhanced upconversion emission for improved near-infrared photoresponse. Nanoscale 6 , 817–824 (2014)	Yin, Z. et al. Remarkable enhancement of upconversion fluorescence and confocal imaging of PMMA Opal/NaYF ₄ :Yb ³⁺ , Tm ³⁺ /Er ³⁺ nanocrystals. ChemComm 49 , 3781–3783 (2013)
Photonic structure	Photonic Structure	SiO ₂ opal	3D PC of monodisperse carboxylate-modified polystyrene spheres	PMMA opal PC
	number of designs investigated	2	3	5
UC material	Upconverter Material	NaYF ₄ :Yb/Er and NaYF ₄ :Yb/Tm	Yb/Er and Yb/Tm co-doped NaYF ₄	NaYF ₄ :Yb ³⁺ , Tm ³⁺ /Er ³⁺
	Form of UC material	UCNPs	UCNPs	UCNPs
	Amount of UC material in structure	~1 μm thin layer on top of opal PC	~ 200 nm thin layer on top of opal PC	one thin layer in spaces around opal colloids on top layer, thickness not reported
	amount limited by PC design	yes	yes	yes
Reference	Reference structure	Thin UC nanoparticle film on glass substrate	Thin UC nanoparticle film on glass substrate	Thin UC nanoparticle film on glass substrate
Excitation	irradiance	-	4 W cm ⁻²	-
	Power	400 mW		5 mW
	wavelength	980 nm	980 nm	980 nm
	Angle (to surface normal)	-	0°	65° - 90° (0° - 25° off surface plane)
Detection	Angle (to surface normal)	-	-	180° transmission
UC enhancement	Wavelength/emission	450 nm / 541 nm	540 nm / 650 nm	"overall UC" of emissions between 400 nm and 800 nm
	Enhancement factor	34 / 23	30 / 30	30
	Spectral width	-	-	-
	Angle width	-	-	-
Simulation	Reflectance/transmittance to determine position of bandgap	yes	yes	yes
	Electric field intensity	-	-	-
	Local density of optical states	-	-	-
	Upconversion dynamics	-	-	-

Table S1 Overview of design, experimental parameters and simulations of photonic structures (PS) for upconversion (UC) enhancement. A selection of reports on opal photonic crystals (PCs). (Further abbreviations: upconverter nanoparticles (UCNPs)).

		Inverse opal PCs	Inverse opal PCs	2D PCs
Publication		Xu, S. et al. NaYF ₄ :Yb,Tm nanocrystals and TiO ₂ inverse opal composite films: a novel device for upconversion enhancement and solid-based sensing of avidin. Nanoscale 6 , 5859–5870 (2014)	Zhang, F., Deng, Y., Shi, Y., Zhang, R. & Zhao, D. Photoluminescence modification in upconversion rare-earth fluoridenanocrystal array constructed photonic crystals. J. Mater. Chem. 20 , 3895–3900 (2010)	Wang, H. et al. Remarkable enhancement of upconversion luminescence on 2-D anodic aluminum oxide photonic crystals. Nanoscale 8 , 10004–10009 (2016)
Photonic structure	Photonic Structure	TiO ₂ inverse opal PCs	polystyrene inverse opal PCs	anodic aluminum oxides two-dimensional photonic crystal
	number of designs investigated	5	3	~25
UC material	Upconverter Material	NaYF ₄ :Yb ³⁺ , Tm ³⁺ (Er ³⁺)	NaYF ₄ :Yb ³⁺ /Er ³⁺	NaYF ₄ :Yb ³⁺ Er ³⁺
	Form of UC material	UCNPs	UCNPs	UCNPs
	Amount of UC material in structure	UCNPs are embedded in the voids of TiO ₂ IOPCs, thickness not reported	inside the voids of the IOPC 1 cm pathlength sample investigated in cuvette	1.3 μm thin film on top of 2D PC
	amount limited by PC design	no	no	yes
Reference	Reference structure	Thin UC nanoparticle film on glass substrate	UCNPs in cuvette	Thin UC nanoparticle film on glass substrate
Excitation	irradiance	48 mW mm ⁻²	-	33 W cm ⁻²
	Power	0.1 - 0.9 W	800 mW	
	wavelength	980 nm	978 nm	980 nm
	Angle (to surface normal)	0°	-	0°
Detection	Angle (to surface normal)	180° transmission	-	~ 45°
UC enhancement	Wavelength/emission	"overall UC" of emissions between 300 nm and 800 nm	515 nm, 565 nm, 640 nm, 675 nm	"overall UC" of emissions of red and green / green / red
	Enhancement factor	43 (decreasing for higher power)	4.6 (same for each emission reported)	65 / 50 / 130
	Spectral width	-	-	-
	Angle width	-	-	-
Simulation	Reflectance/transmittance to determine position of bandgap		yes	yes
	Electric field intensity	-	-	yes
	Local density of optical states	-	-	-
	Upconversion dynamics	-	-	-

Table S2 Overview of design, experimental parameters and simulations of photonic structures (PS) for upconversion (UC) enhancement. A selection of reports on inverse opal photonic crystals (PCs) and 2D PCs. (Further abbreviations: upconverter nanoparticles (UCNPs)).

		Waveguides	Cavities	
Publication		Lin, J. H. et al. Giant Enhancement of Upconversion Fluorescence of NaYF ₄ :Yb ³⁺ ,Tm ³⁺ Nanocrystals with Resonant Waveguide Grating Substrate. ACS Photonics 2 , 530–536 (2015)	Rojas-Hernandez, R. E., Santos, L. F. & Almeida, R. M. Photonic crystal assisted up-converter based on Tb ³⁺ / Yb ³⁺ - Doped aluminosilicate glass. Opt. Mater. 83 , 61–67 (2018)	Yang, J., Li, A.-H., Chen, C. & Sun, Z. Cavity controlled upconversion luminescence in Ag-capped NaYF ₄ :Yb,Er micron rod. J. Lumin. 187 , 466–470 (2017)
Photonic structure	Photonic Structure	waveguide structure	microcavity structure consisting of 21 layers: two Bragg reflectors of alternating TiO ₂ and Tb ³⁺ /Yb ³⁺ - doped aluminosilicate glass, separated by Tb ³⁺ /Yb ³⁺ - doped aluminosilicate glass defect layer	Ag-capped β-NaYF ₄ :Yb,Er micron rods on a PDMS cavity
	number of designs investigated	1	9	32
UC material	Upconverter Material	NaYF ₄ :Yb ³⁺ ,Tm ³⁺	Tb ³⁺ /Yb ³⁺	β-NaYF ₄ :Yb,Er
	Form of UC material	UCNPs	UC doped aluminosilicate glass	UC micron rods
	Amount of UC material in structure	~200 nm thin layer on top of waveguide structure	11 layers with a summed up thickness of ~ 1 μm	micron rods length 8 μm , diamter 1.8 μm
	amount limited by PC design	yes	no	yes
Reference	Reference structure	Nonpatterned area of the sample	Layer stack of only the Tb ³⁺ /Yb ³⁺ -doped aluminosilicate layers. Seperate reference for each microcavity design	i) Bare rods on glass and ii) 200 nm Ag film on glass and bare rods on top
Excitation	irradiance	65 W cm ⁻² (scanned ~10-70 W cm ⁻²)	-	500 W cm ⁻²
	Power		2 W (scan 1 W to 4 W)	
	wavelength	976 nm	975 nm	975 nm
	Angle (to surface normal)	0° - 50°		0°
Detection	Angle (to surface normal)	0° - 50°	one distinct detection angle (not given which)	0° (using beam splitting)
UC enhancement	Wavelength/emission	450nm / 480 nm / 650 nm	green	to reference i Including statistics of red and green / max for green / max for red / to refence ii
	Enhancement factor	6.8*10 ⁴ / 8.8*10 ⁴ / 1.6*10 ⁴	25	4 / 5.2 / 5.0 / 1.8
	Spectral width	Enhancement occurs only at very specific excitation wavelengths	-	-
	Angle width	Extremely narrow excitation angle range of maximally 0.75° (half angle) is enhanced. UC enhancement on center at 31.5° is 10 ⁴ , off center at 30.75° it drops by 3 orders of magnitude to 1.4.	-	-
Simulation	Reflectance/transmittance to determine position of bandgap	yes	yes	yes
	Electric field intensity	yes	-	yes
	Local density of optical states	-	-	FDTD simulations of electric dipole emission and Purcell factor
	Upconversion dynamics	-	-	-

Table S3 Overview of design, experimental parameters and simulations of photonic structures (PS) for upconversion (UC) enhancement. A selection of reports on waveguide and cavity structures. (Further abbreviations: upconverter nanoparticles (UCNPs)).

		Multilayer stacks	Multilayer stacks
Publication		Johnson, C. M., Reece, P. J. & Conibeer, G. J. Theoretical and experimental evaluation of silicon photonic structures for enhanced erbium up-conversion luminescence. Sol. Energy Mater. Sol. Cells 112 , 168–181 (2013)	Hofmann et al, Upconversion enhancement in 1D-photonic crystals: bringing together theory and experiment, manuscript under consideration (2020)
Photonic structure	Photonic Structure	multilayer stack of ~60 layers of Er ³⁺ -doped porous silicon with alternating refractive index (both, high and low refractive index layers are doped)	Bragg structure made of 4 active layers of PMMA with embedded UCNPs and 5 surrounding TiO ₂ layers
	number of designs investigated	1	40
UC material	Upconverter Material	Er ³⁺	NaYF ₄ :25%Er ³⁺
	Form of UC material	UC doped porous silicon	UCNPs
	Amount of UC material in structure	thin layers with ~ 15 μm summed up thickness	~1.2 μm summed up thickness of all UC layers
	amount limited by PC design	no	no
Reference	Reference structure	None. The enhancement is calculated relative to the lowest measured UC emission at 38°	Layer stack of only the UC layers on glass to gain the same total thickness as the UC layers in each Bragg structure
Excitation	irradiance	-	max at 1.48 W cm ⁻² (scan ~ 0.18 W cm ⁻² - 1.5 W cm ⁻²)
	Power	200 mW	1 mW - 10 mW
	wavelength	1550 nm	1523 nm for max UC enh. 1500 nm - 1560 scan
	Angle (to surface normal)	max UC enhancement at 34°, scanned: 21° - 38°	4° for max, and scanned 0° - 75°
Detection	Angle (to surface normal)	0°	Integrating sphere
UC enhancement	Wavelength/emission	980 nm / green	984 nm / 814 nm
	Enhancement factor	5 / 26.5	max 4.1 (mean 2.4) / ~5
	Spectral width	-	~ 60 nm
	Angle width	~ 4° (full angle) max UC enhancement at 34° incident angle, dropping by about a factor of 5 within 2° to lower and higher angles)	~ 30° half angle
Simulation	Reflectance/transmittance to determine position of bandgap	yes	yes
	Electric field intensity	-	yes
	Local density of optical states	-	MIT Photonic bands simulation and histogramming method to determine local density of optical states
	Upconversion dynamics	-	rate equation model of UC dynamics including the impact of photonic effects and production accuracy of the photonic structure

Table S4 Overview of design, experimental parameters and simulations of photonic structures (PS) for upconversion (UC) enhancement. A selection of reports on multilayer stacks. (Further abbreviations: upconverter nanoparticles (UCNPs))

REVIEWERS' COMMENTS

Reviewer #1 (Remarks to the Author):

The most recent response from the authors provides a detailed comparison with prior work on upconversion in a new table. As such, this addresses the follow-up points 1 and 2 raised in the most recent re-review. The only shortcoming still remaining is extremely minor, in that there is a typo in the spelling of enhancement. Follow-up point 3 has also been addressed in a clear and satisfactory manner. Therefore, I would recommend that this work be published after performing a final proofing check. I trust that the authors can take care of this, so no further reviews will be needed. Thank you very much.

In this final resubmission, we would like to thank again all three reviewers for all their thoughtful comments and questions. Based on these questions, we were able to significantly improve our manuscript in the course of the review process. Working out the contribution of this publication to a wider community and sorting out what needed to be elaborated in more detail to clarify this point, was very helpful for us.

The *reviewer's comments and questions* are marked as italic text.

Reviewer #1 (Remarks to the Author):

The most recent response from the authors provides a detailed comparison with prior work on upconversion in a new table. As such, this addresses the follow-up points 1 and 2 raised in the most recent re-review. The only shortcoming still remaining is extremely minor, in that there is a typo in the spelling of enhancement. Follow-up point 3 has also been addressed in a clear and satisfactory manner. Therefore, I would recommend that this work be published after performing a final proofing check. I trust that the authors can take care of this, so no further reviews will be needed. Thank you very much.

We would like to thank very much you for this final positive feedback. And thank you for pointing out the spelling mistake, we corrected it in the final submitted version and performed a final proofing check on writing, graphs and tables.